# THE LOTTERY TICKET HYPOTHESIS: FINDING SPARSE, TRAINABLE NEURAL NETWORKS

**Jonathan Frankle**
MIT CSAIL
jfrankle@csail.mit.edu

**Michael Carbin**
MIT CSAIL
mcarbin@csail.mit.edu

## ABSTRACT

Neural network pruning techniques can reduce the parameter counts of trained networks by over 90%, decreasing storage requirements and improving computational performance of inference without compromising accuracy. However, contemporary experience is that the sparse architectures produced by pruning are difficult to train from the start, which would similarly improve training performance.

We find that a standard pruning technique naturally uncovers subnetworks whose initializations made them capable of training effectively. Based on these results, we articulate the *lottery ticket hypothesis*: dense, randomly-initialized, feed-forward networks contain subnetworks (*winning tickets*) that—when trained in isolation—reach test accuracy comparable to the original network in a similar number of iterations. The winning tickets we find have won the initialization lottery: their connections have initial weights that make training particularly effective.

We present an algorithm to identify winning tickets and a series of experiments that support the lottery ticket hypothesis and the importance of these fortuitous initializations. We consistently find winning tickets that are less than 10-20% of the size of several fully-connected and convolutional feed-forward architectures for MNIST and CIFAR10. Above this size, the winning tickets that we find learn faster than the original network and reach higher test accuracy.

## 1 INTRODUCTION

Techniques for eliminating unnecessary weights from neural networks (*pruning*) (LeCun et al., 1990; Hassibi & Stork, 1993; Han et al., 2015; Li et al., 2016) can reduce parameter-counts by more than 90% without harming accuracy. Doing so decreases the size (Han et al., 2015; Hinton et al., 2015) or energy consumption (Yang et al., 2017; Molchanov et al., 2016; Luo et al., 2017) of the trained networks, making inference more efficient. However, if a network can be reduced in size, why do we not train this smaller architecture instead in the interest of making training more efficient as well? Contemporary experience is that the architectures uncovered by pruning are harder to train from the start, reaching lower accuracy than the original networks.[1]

Consider an example. In Figure 1, we randomly sample and train subnetworks from a fully-connected network for MNIST and convolutional networks for CIFAR10. Random sampling models the effect of the unstructured pruning used by LeCun et al. (1990) and Han et al. (2015). Across various levels of sparsity, dashed lines trace the iteration of minimum validation loss[2] and the test accuracy at that iteration. The sparser the network, the slower the learning and the lower the eventual test accuracy.

---

[1] "Training a pruned model from scratch performs worse than retraining a pruned model, which may indicate the difficulty of training a network with a small capacity." (Li et al., 2016) "During retraining, it is better to retain the weights from the initial training phase for the connections that survived pruning than it is to re-initialize the pruned layers...gradient descent is able to find a good solution when the network is initially trained, but not after re-initializing some layers and retraining them." (Han et al., 2015)

[2] As a proxy for the speed at which a network learns, we use the iteration at which an early-stopping criterion would end training. The particular early-stopping criterion we employ throughout this paper is the iteration of minimum validation loss during training. See Appendix C for more details on this choice.

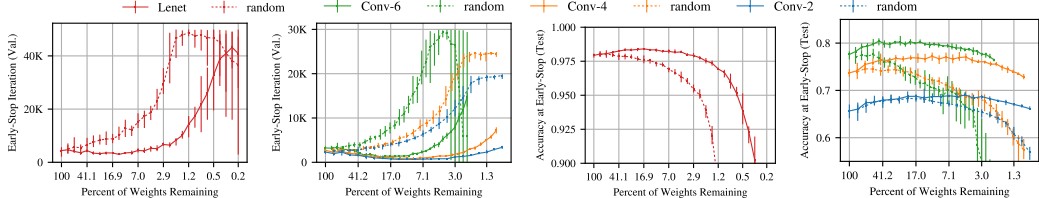

Figure 1: The iteration at which early-stopping would occur (left) and the test accuracy at that iteration (right) of the Lenet architecture for MNIST and the Conv-2, Conv-4, and Conv-6 architectures for CIFAR10 (see Figure 2) when trained starting at various sizes. Dashed lines are randomly sampled sparse networks (average of ten trials). Solid lines are winning tickets (average of five trials).

In this paper, we show that there consistently exist smaller subnetworks that train from the start and learn at least as fast as their larger counterparts while reaching similar test accuracy. Solid lines in Figure 1 show networks that we find. Based on these results, we state *the lottery ticket hypothesis*.

**The Lottery Ticket Hypothesis.** *A randomly-initialized, dense neural network contains a subnetwork that is initialized such that—when trained in isolation—it can match the test accuracy of the original network after training for at most the same number of iterations.*

More formally, consider a dense feed-forward neural network $f(x; \theta)$ with initial parameters $\theta = \theta_0 \sim \mathcal{D}_\theta$. When optimizing with stochastic gradient descent (SGD) on a training set, $f$ reaches minimum validation loss $l$ at iteration $j$ with test accuracy $a$. In addition, consider training $f(x; m \odot \theta)$ with a mask $m \in \{0, 1\}^{|\theta|}$ on its parameters such that its initialization is $m \odot \theta_0$. When optimizing with SGD on the same training set (with $m$ fixed), $f$ reaches minimum validation loss $l'$ at iteration $j'$ with test accuracy $a'$. The lottery ticket hypothesis predicts that $\exists\, m$ for which $j' \leq j$ (*commensurate training time*), $a' \geq a$ (*commensurate accuracy*), and $\|m\|_0 \ll |\theta|$ (*fewer parameters*).

We find that a standard pruning technique automatically uncovers such trainable subnetworks from fully-connected and convolutional feed-forward networks. We designate these trainable subnetworks, $f(x; m \odot \theta_0)$, *winning tickets*, since those that we find have won the initialization lottery with a combination of weights and connections capable of learning. When their parameters are randomly reinitialized ($f(x; m \odot \theta_0')$ where $\theta_0' \sim \mathcal{D}_\theta$), our winning tickets no longer match the performance of the original network, offering evidence that these smaller networks do not train effectively unless they are appropriately initialized.

**Identifying winning tickets.** We identify a winning ticket by training a network and pruning its smallest-magnitude weights. The remaining, unpruned connections constitute the architecture of the winning ticket. Unique to our work, each unpruned connection's value is then reset to its initialization from original network *before* it was trained. This forms our central experiment:

1. Randomly initialize a neural network $f(x; \theta_0)$ (where $\theta_0 \sim \mathcal{D}_\theta$).
2. Train the network for $j$ iterations, arriving at parameters $\theta_j$.
3. Prune $p\%$ of the parameters in $\theta_j$, creating a mask $m$.
4. Reset the remaining parameters to their values in $\theta_0$, creating the winning ticket $f(x; m \odot \theta_0)$.

As described, this pruning approach is *one-shot*: the network is trained once, $p\%$ of weights are pruned, and the surviving weights are reset. However, in this paper, we focus on *iterative pruning*, which repeatedly trains, prunes, and resets the network over $n$ rounds; each round prunes $p^{\frac{1}{n}}\%$ of the weights that survive the previous round. Our results show that iterative pruning finds winning tickets that match the accuracy of the original network at smaller sizes than does one-shot pruning.

**Results.** We identify winning tickets in a fully-connected architecture for MNIST and convolutional architectures for CIFAR10 across several optimization strategies (SGD, momentum, and Adam) with techniques like dropout, weight decay, batchnorm, and residual connections. We use an unstructured pruning technique, so these winning tickets are sparse. In deeper networks, our pruning-based strategy for finding winning tickets is sensitive to the learning rate: it requires warmup to find winning tickets at higher learning rates. The winning tickets we find are 10-20% (or less) of the size of the original

| Network | Lenet | Conv-2 | Conv-4 | Conv-6 | Resnet-18 | VGG-19 |
|---|---|---|---|---|---|---|
| Convolutions | | 64, 64, pool | 64, 64, pool 128, 128, pool | 64, 64, pool 128, 128, pool 256, 256, pool | 16, 3x[16, 16] 3x[32, 32] 3x[64, 64] | 2x64 pool 2x128 pool, 4x256, pool 4x512, pool, 4x512 |
| FC Layers | 300, 100, 10 | 256, 256, 10 | 256, 256, 10 | 256, 256, 10 | avg-pool, 10 | avg-pool, 10 |
| All/Conv Weights | 266K | 4.3M / 38K | 2.4M / 260K | 1.7M / 1.1M | 274K / 270K | 20.0M |
| Iterations/Batch | 50K / 60 | 20K / 60 | 25K / 60 | 30K / 60 | 30K / 128 | 112K / 64 |
| Optimizer | Adam 1.2e-3 | Adam 2e-4 | Adam 3e-4 | Adam 3e-4 | ← SGD 0.1-0.01-0.001 Momentum 0.9 → | |
| Pruning Rate | fc20% | conv10% fc20% | conv10% fc20% | conv15% fc20% | conv20% fc0% | conv20% fc0% |

Figure 2: Architectures tested in this paper. Convolutions are 3x3. Lenet is from LeCun et al. (1998). Conv-2/4/6 are variants of VGG (Simonyan & Zisserman, 2014). Resnet-18 is from He et al. (2016). VGG-19 for CIFAR10 is adapted from Liu et al. (2019). Initializations are Gaussian Glorot (Glorot & Bengio, 2010). Brackets denote residual connections around layers.

network (*smaller size*). Down to that size, they meet or exceed the original network's test accuracy (*commensurate accuracy*) in at most the same number of iterations (*commensurate training time*). When randomly reinitialized, winning tickets perform far worse, meaning structure alone cannot explain a winning ticket's success.

**The Lottery Ticket Conjecture.** Returning to our motivating question, we extend our hypothesis into an untested conjecture that SGD seeks out and trains a subset of well-initialized weights. Dense, randomly-initialized networks are easier to train than the sparse networks that result from pruning because there are more possible subnetworks from which training might recover a winning ticket.

**Contributions.**

- We demonstrate that pruning uncovers trainable subnetworks that reach test accuracy comparable to the original networks from which they derived in a comparable number of iterations.

- We show that pruning finds winning tickets that learn faster than the original network while reaching higher test accuracy and generalizing better.

- We propose the *lottery ticket hypothesis* as a new perspective on the composition of neural networks to explain these findings.

**Implications.** In this paper, we empirically study the lottery ticket hypothesis. Now that we have demonstrated the existence of winning tickets, we hope to exploit this knowledge to:

*Improve training performance.* Since winning tickets can be trained from the start in isolation, a hope is that we can design training schemes that search for winning tickets and prune as early as possible.

*Design better networks.* Winning tickets reveal combinations of sparse architectures and initializations that are particularly adept at learning. We can take inspiration from winning tickets to design new architectures and initialization schemes with the same properties that are conducive to learning. We may even be able to transfer winning tickets discovered for one task to many others.

*Improve our theoretical understanding of neural networks.* We can study why randomly-initialized feed-forward networks seem to contain winning tickets and potential implications for theoretical study of optimization (Du et al., 2019) and generalization (Zhou et al., 2018; Arora et al., 2018).

## 2 WINNING TICKETS IN FULLY-CONNECTED NETWORKS

In this Section, we assess the lottery ticket hypothesis as applied to fully-connected networks trained on MNIST. We use the Lenet-300-100 architecture (LeCun et al., 1998) as described in Figure 2. We follow the outline from Section 1: after randomly initializing and training a network, we prune the network and reset the remaining connections to their original initializations. We use a simple layer-wise pruning heuristic: remove a percentage of the weights with the lowest magnitudes within each layer (as in Han et al. (2015)). Connections to outputs are pruned at half of the rate of the rest of the network. We explore other hyperparameters in Appendix G, including learning rates, optimization strategies (SGD, momentum), initialization schemes, and network sizes.

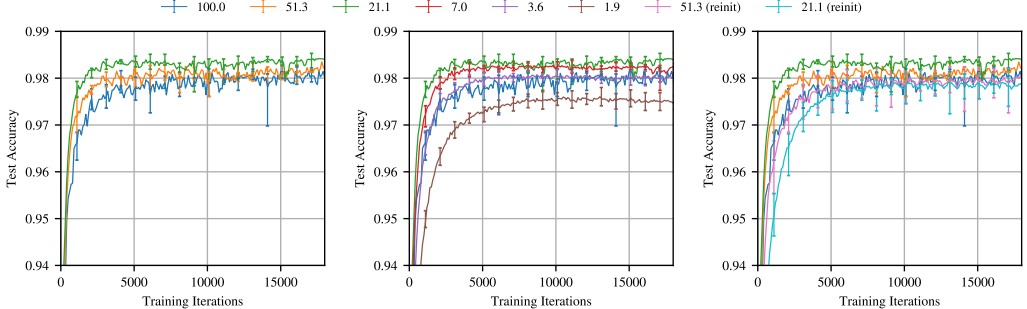

Figure 3: Test accuracy on Lenet (iterative pruning) as training proceeds. Each curve is the average of five trials. Labels are $P_m$—the fraction of weights remaining in the network after pruning. Error bars are the minimum and maximum of any trial.

**Notation.** $P_m = \frac{\|m\|_0}{|\theta|}$ is the sparsity of mask $m$, e.g., $P_m = 25\%$ when 75% of weights are pruned.

**Iterative pruning.** The winning tickets we find learn faster than the original network. Figure 3 plots the average test accuracy when training winning tickets iteratively pruned to various extents. Error bars are the minimum and maximum of five runs. For the first pruning rounds, networks learn faster and reach higher test accuracy the more they are pruned (left graph in Figure 3). A winning ticket comprising 51.3% of the weights from the original network (i.e., $P_m = 51.3\%$) reaches higher test accuracy faster than the original network but slower than when $P_m = 21.1\%$. When $P_m < 21.1\%$, learning slows (middle graph). When $P_m = 3.6\%$, a winning ticket regresses to the performance of the original network. A similar pattern repeats throughout this paper.

Figure 4a summarizes this behavior for all pruning levels when iteratively pruning by 20% per iteration (blue). On the left is the iteration at which each network reaches minimum validation loss (i.e., when the early-stopping criterion would halt training) in relation to the percent of weights remaining after pruning; in the middle is test accuracy at that iteration. We use the iteration at which the early-stopping criterion is met as a proxy for how quickly the network learns.

The winning tickets learn faster as $P_m$ decreases from 100% to 21%, at which point early-stopping occurs 38% earlier than for the original network. Further pruning causes learning to slow, returning to the early-stopping performance of the original network when $P_m = 3.6\%$. Test accuracy increases with pruning, improving by more than 0.3 percentage points when $P_m = 13.5\%$; after this point, accuracy decreases, returning to the level of the original network when $P_m = 3.6\%$.

At early stopping, training accuracy (Figure 4a, right) increases with pruning in a similar pattern to test accuracy, seemingly implying that winning tickets optimize more effectively but do not generalize better. However, at iteration 50,000 (Figure 4b), iteratively-pruned winning tickets still see a test accuracy improvement of up to 0.35 percentage points in spite of the fact that training accuracy reaches 100% for nearly all networks (Appendix D, Figure 12). This means that the gap between training accuracy and test accuracy is smaller for winning tickets, pointing to improved generalization.

**Random reinitialization.** To measure the importance of a winning ticket's initialization, we retain the structure of a winning ticket (i.e., the mask $m$) but randomly sample a new initialization $\theta_0' \sim \mathcal{D}_\theta$. We randomly reinitialize each winning ticket three times, making 15 total per point in Figure 4. We find that initialization is crucial for the efficacy of a winning ticket. The right graph in Figure 3 shows this experiment for iterative pruning. In addition to the original network and winning tickets at $P_m = 51\%$ and 21% are the random reinitialization experiments. Where the winning tickets learn faster as they are pruned, they learn progressively slower when randomly reinitialized.

The broader results of this experiment are orange line in Figure 4a. Unlike winning tickets, the reinitialized networks learn increasingly slower than the original network and lose test accuracy after little pruning. The average reinitialized iterative winning ticket's test accuracy drops off from the original accuracy when $P_m = 21.1\%$, compared to 2.9% for the winning ticket. When $P_m = 21\%$, the winning ticket reaches minimum validation loss 2.51x faster than when reinitialized and is half a percentage point more accurate. All networks reach 100% training accuracy for $P_m \geq 5\%$; Figure

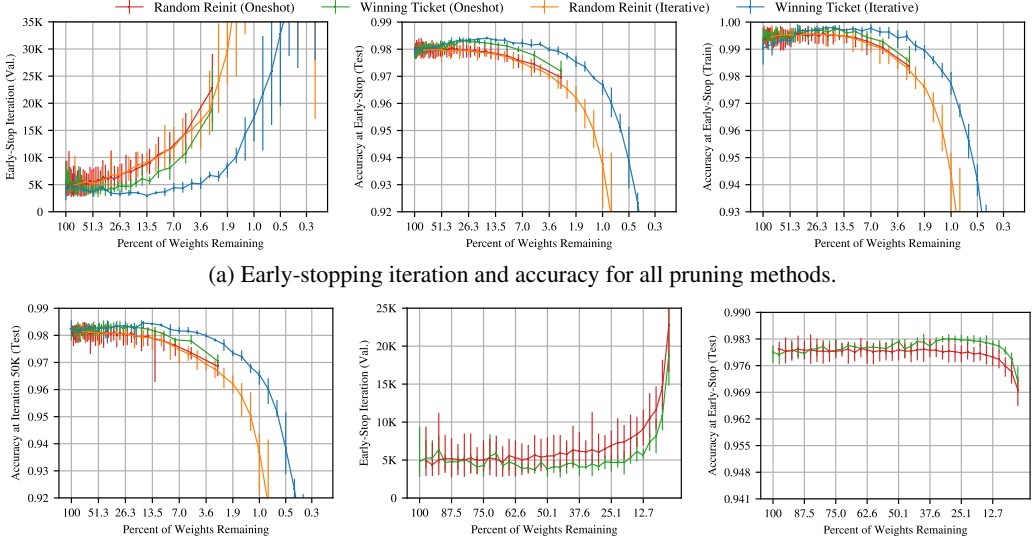

(a) Early-stopping iteration and accuracy for all pruning methods.

(b) Accuracy at end of training.  (c) Early-stopping iteration and accuracy for one-shot pruning.

Figure 4: Early-stopping iteration and accuracy of Lenet under one-shot and iterative pruning. Average of five trials; error bars for the minimum and maximum values. At iteration 50,000, training accuracy $\approx 100\%$ for $P_m \geq 2\%$ for iterative winning tickets (see Appendix D, Figure 12).

4b therefore shows that the winning tickets generalize substantially better than when randomly reinitialized. This experiment supports the lottery ticket hypothesis' emphasis on initialization: the original initialization withstands and benefits from pruning, while the random reinitialization's performance immediately suffers and diminishes steadily.

**One-shot pruning.** Although iterative pruning extracts smaller winning tickets, repeated training means they are costly to find. One-shot pruning makes it possible to identify winning tickets without this repeated training. Figure 4c shows the results of one-shot pruning (green) and randomly reinitializing (red); one-shot pruning does indeed find winning tickets. When $67.5\% \geq P_m \geq 17.6\%$, the average winning tickets reach minimum validation accuracy earlier than the original network. When $95.0\% \geq P_m \geq 5.17\%$, test accuracy is higher than the original network. However, iteratively-pruned winning tickets learn faster and reach higher test accuracy at smaller network sizes. The green and red lines in Figure 4c are reproduced on the logarithmic axes of Figure 4a, making this performance gap clear. Since our goal is to identify the smallest possible winning tickets, we focus on iterative pruning throughout the rest of the paper.

## 3   WINNING TICKETS IN CONVOLUTIONAL NETWORKS

Here, we apply the lottery ticket hypothesis to convolutional networks on CIFAR10, increasing both the complexity of the learning problem and the size of the networks. We consider the Conv-2, Conv-4, and Conv-6 architectures in Figure 2, which are scaled-down variants of the VGG (Simonyan & Zisserman, 2014) family. The networks have two, four, or six convolutional layers followed by two fully-connected layers; max-pooling occurs after every two convolutional layers. The networks cover a range from near-fully-connected to traditional convolutional networks, with less than 1% of parameters in convolutional layers in Conv-2 to nearly two thirds in Conv-6.[3]

**Finding winning tickets.** The solid lines in Figure 5 (top) show the iterative lottery ticket experiment on Conv-2 (blue), Conv-4 (orange), and Conv-6 (green) at the per-layer pruning rates from Figure 2. The pattern from Lenet in Section 2 repeats: as the network is pruned, it learns faster and test accuracy rises as compared to the original network. In this case, the results are more pronounced. Winning

---

[3]Appendix H explores other hyperparameters, including learning rates, optimization strategies (SGD, momentum), and the relative rates at which to prune convolutional and fully-connected layers.

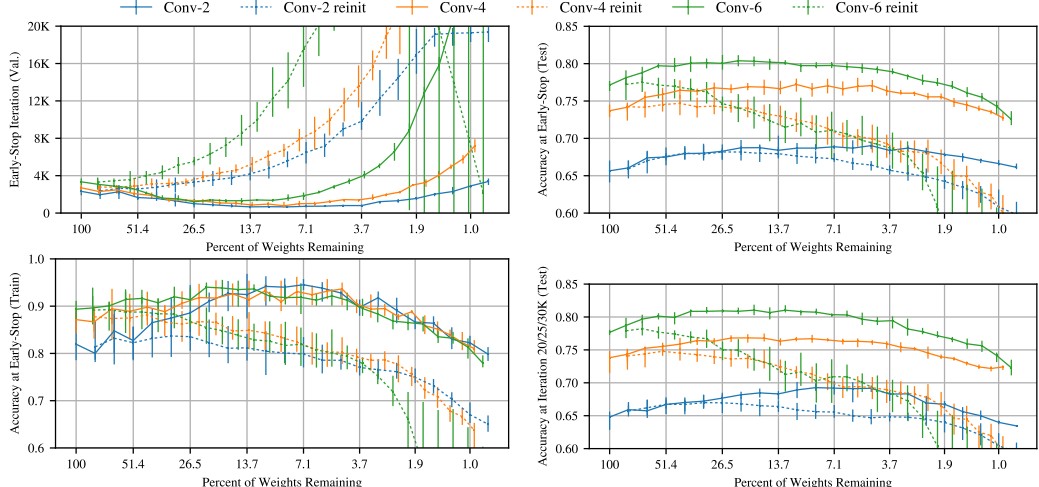

Figure 5: Early-stopping iteration and test and training accuracy of the Conv-2/4/6 architectures when iteratively pruned and when randomly reinitialized. Each solid line is the average of five trials; each dashed line is the average of fifteen reinitializations (three per trial). The bottom right graph plots test accuracy of winning tickets at iterations corresponding to the last iteration of training for the original network (20,000 for Conv-2, 25,000 for Conv-4, and 30,000 for Conv-6); at this iteration, training accuracy $\approx 100\%$ for $P_m \geq 2\%$ for winning tickets (see Appendix D).

tickets reach minimum validation loss at best 3.5x faster for Conv-2 ($P_m = 8.8\%$), 3.5x for Conv-4 ($P_m = 9.2\%$), and 2.5x for Conv-6 ($P_m = 15.1\%$). Test accuracy improves at best 3.4 percentage points for Conv-2 ($P_m = 4.6\%$), 3.5 for Conv-4 ($P_m = 11.1\%$), and 3.3 for Conv-6 ($P_m = 26.4\%$). All three networks remain above their original average test accuracy when $P_m > 2\%$.

As in Section 2, training accuracy at the early-stopping iteration rises with test accuracy. However, at iteration 20,000 for Conv-2, 25,000 for Conv-4, and 30,000 for Conv-6 (the iterations corresponding to the final training iteration for the original network), training accuracy reaches 100% for all networks when $P_m \geq 2\%$ (Appendix D, Figure 13) and winning tickets still maintain higher test accuracy (Figure 5 bottom right). This means that the gap between test and training accuracy is smaller for winning tickets, indicating they generalize better.

**Random reinitialization.** We repeat the random reinitialization experiment from Section 2, which appears as the dashed lines in Figure 5. These networks again take increasingly longer to learn upon continued pruning. Just as with Lenet on MNIST (Section 2), test accuracy drops off more quickly for the random reinitialization experiments. However, unlike Lenet, test accuracy at early-stopping time initially remains steady and even improves for Conv-2 and Conv-4, indicating that—at moderate levels of pruning—the structure of the winning tickets alone may lead to better accuracy.

**Dropout.** Dropout (Srivastava et al., 2014; Hinton et al., 2012) improves accuracy by randomly disabling a fraction of the units (i.e., randomly sampling a subnetwork) on each training iteration. Baldi & Sadowski (2013) characterize dropout as simultaneously training the ensemble of all subnetworks. Since the lottery ticket hypothesis suggests that one of these subnetworks comprises a winning ticket, it is natural to ask whether dropout and our strategy for finding winning tickets interact.

Figure 6 shows the results of training Conv-2, Conv-4, and Conv-6 with a dropout rate of 0.5. Dashed lines are the network performance without dropout (the solid lines in Figure 5).[4] We continue to find winning tickets when training with dropout. Dropout increases initial test accuracy (2.1, 3.0, and 2.4 percentage points on average for Conv-2, Conv-4, and Conv-6, respectively), and iterative pruning increases it further (up to an additional 2.3, 4.6, and 4.7 percentage points, respectively, on average). Learning becomes faster with iterative pruning as before, but less dramatically in the case of Conv-2.

---

[4]We choose new learning rates for the networks as trained with dropout—see Appendix H.5.

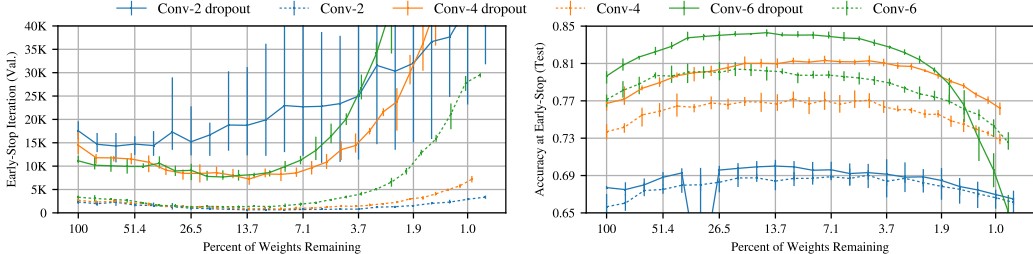

Figure 6: Early-stopping iteration and test accuracy at early-stopping of Conv-2/4/6 when iteratively pruned and trained with dropout. The dashed lines are the same networks trained without dropout (the solid lines in Figure 5). Learning rates are 0.0003 for Conv-2 and 0.0002 for Conv-4 and Conv-6.

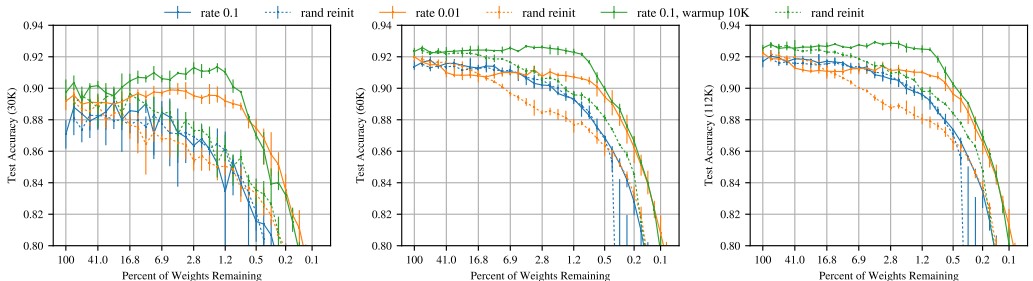

Figure 7: Test accuracy (at 30K, 60K, and 112K iterations) of VGG-19 when iteratively pruned.

These improvements suggest that our iterative pruning strategy interacts with dropout in a complementary way. Srivastava et al. (2014) observe that dropout induces sparse activations in the final network; it is possible that dropout-induced sparsity primes a network to be pruned. If so, dropout techniques that target weights (Wan et al., 2013) or learn per-weight dropout probabilities (Molchanov et al., 2017; Louizos et al., 2018) could make winning tickets even easier to find.

## 4    VGG AND RESNET FOR CIFAR10

Here, we study the lottery ticket hypothesis on networks evocative of the architectures and techniques used in practice. Specifically, we consider VGG-style deep convolutional networks (VGG-19 on CIFAR10—Simonyan & Zisserman (2014)) and residual networks (Resnet-18 on CIFAR10—He et al. (2016)).[5] These networks are trained with batchnorm, weight decay, decreasing learning rate schedules, and augmented training data. We continue to find winning tickets for all of these architectures; however, our method for finding them, iterative pruning, is sensitive to the particular learning rate used. In these experiments, rather than measure early-stopping time (which, for these larger networks, is entangled with learning rate schedules), we plot accuracy at several moments during training to illustrate the relative rates at which accuracy improves.

**Global pruning.** On Lenet and Conv-2/4/6, we prune each layer separately at the same rate. For Resnet-18 and VGG-19, we modify this strategy slightly: we prune these deeper networks *globally*, removing the lowest-magnitude weights collectively across all convolutional layers. In Appendix I.1, we find that global pruning identifies smaller winning tickets for Resnet-18 and VGG-19. Our conjectured explanation for this behavior is as follows: For these deeper networks, some layers have far more parameters than others. For example, the first two convolutional layers of VGG-19 have 1728 and 36864 parameters, while the last has 2.35 million. When all layers are pruned at the same rate, these smaller layers become bottlenecks, preventing us from identifying the smallest possible winning tickets. Global pruning makes it possible to avoid this pitfall.

**VGG-19.** We study the variant VGG-19 adapted for CIFAR10 by Liu et al. (2019); we use the the same training regime and hyperparameters: 160 epochs (112,480 iterations) with SGD with

---

[5]See Figure 2 and Appendices I for details on the networks, hyperparameters, and training regimes.

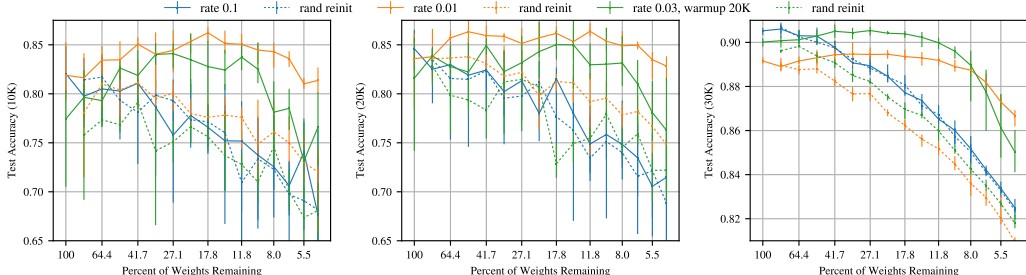

Figure 8: Test accuracy (at 10K, 20K, and 30K iterations) of Resnet-18 when iteratively pruned.

momentum (0.9) and decreasing the learning rate by a factor of 10 at 80 and 120 epochs. This network has 20 million parameters. Figure 7 shows the results of iterative pruning and random reinitialization on VGG-19 at two initial learning rates: 0.1 (used in Liu et al. (2019)) and 0.01. At the higher learning rate, iterative pruning does not find winning tickets, and performance is no better than when the pruned networks are randomly reinitialized. However, at the lower learning rate, the usual pattern reemerges, with subnetworks that remain within 1 percentage point of the original accuracy while $P_m \geq 3.5\%$. (They are not winning tickets, since they do not match the original accuracy.) When randomly reinitialized, the subnetworks lose accuracy as they are pruned in the same manner as other experiments throughout this paper. Although these subnetworks learn faster than the unpruned network early in training (Figure 7 left), this accuracy advantage erodes later in training due to the lower initial learning rate. However, these subnetworks still learn faster than when reinitialized.

To bridge the gap between the lottery ticket behavior of the lower learning rate and the accuracy advantage of the higher learning rate, we explore the effect of linear learning rate warmup from 0 to the initial learning rate over $k$ iterations. Training VGG-19 with warmup ($k = 10000$, green line) at learning rate 0.1 improves the test accuracy of the unpruned network by about one percentage point. Warmup makes it possible to find winning tickets, exceeding this initial accuracy when $P_m \geq 1.5\%$.

**Resnet-18.** Resnet-18 (He et al., 2016) is a 20 layer convolutional network with residual connections designed for CIFAR10. It has 271,000 parameters. We train the network for 30,000 iterations with SGD with momentum (0.9), decreasing the learning rate by a factor of 10 at 20,000 and 25,000 iterations. Figure 8 shows the results of iterative pruning and random reinitialization at learning rates 0.1 (used in He et al. (2016)) and 0.01. These results largely mirror those of VGG: iterative pruning finds winning tickets at the lower learning rate but not the higher learning rate. The accuracy of the best winning tickets at the lower learning rate (89.5% when $41.7\% \geq P_m \geq 21.9\%$) falls short of the original network's accuracy at the higher learning rate (90.5%). At lower learning rate, the winning ticket again initially learns faster (left plots of Figure 8), but falls behind the unpruned network at the higher learning rate later in training (right plot). Winning tickets trained with warmup close the accuracy gap with the unpruned network at the higher learning rate, reaching 90.5% test accuracy with learning rate 0.03 (warmup, $k = 20000$) at $P_m = 27.1\%$. For these hyperparameters, we still find winning tickets when $P_m \geq 11.8\%$. Even with warmup, however, we could not find hyperparameters for which we could identify winning tickets at the original learning rate, 0.1.

## 5 DISCUSSION

Existing work on neural network pruning (e.g., Han et al. (2015)) demonstrates that the function learned by a neural network can often be represented with fewer parameters. Pruning typically proceeds by training the original network, removing connections, and further fine-tuning. In effect, the initial training initializes the weights of the pruned network so that it can learn in isolation during fine-tuning. We seek to determine if similarly sparse networks can learn from the start. We find that the architectures studied in this paper reliably contain such trainable subnetworks, and the lottery ticket hypothesis proposes that this property applies in general. Our empirical study of the existence and nature of winning tickets invites a number of follow-up questions.

**The importance of winning ticket initialization.** When randomly reinitialized, a winning ticket learns more slowly and achieves lower test accuracy, suggesting that initialization is important to its success. One possible explanation for this behavior is these initial weights are close to their final

values after training—that in the most extreme case, they are already trained. However, experiments in Appendix F show the opposite—that the winning ticket weights move further than other weights. This suggests that the benefit of the initialization is connected to the optimization algorithm, dataset, and model. For example, the winning ticket initialization might land in a region of the loss landscape that is particularly amenable to optimization by the chosen optimization algorithm.

Liu et al. (2019) find that pruned networks are indeed trainable when randomly reinitialized, seemingly contradicting conventional wisdom and our random reinitialization experiments. For example, on VGG-19 (for which we share the same setup), they find that networks pruned by up to 80% and randomly reinitialized match the accuracy of the original network. Our experiments in Figure 7 confirm these findings at this level of sparsity (below which Liu et al. do not present data). However, after further pruning, initialization matters: we find winning tickets when VGG-19 is pruned by up to 98.5%; when reinitialized, these tickets reach much lower accuracy. We hypothesize that—up to a certain level of sparsity—highly overparameterized networks can be pruned, reinitialized, and retrained successfully; however, beyond this point, extremely pruned, less severely overparamterized networks only maintain accuracy with fortuitous initialization.

**The importance of winning ticket structure.** The initialization that gives rise to a winning ticket is arranged in a particular sparse architecture. Since we uncover winning tickets through heavy use of training data, we hypothesize that the structure of our winning tickets encodes an inductive bias customized to the learning task at hand. Cohen & Shashua (2016) show that the inductive bias embedded in the structure of a deep network determines the kinds of data that it can separate more parameter-efficiently than can a shallow network; although Cohen & Shashua (2016) focus on the pooling geometry of convolutional networks, a similar effect may be at play with the structure of winning tickets, allowing them to learn even when heavily pruned.

**The improved generalization of winning tickets.** We reliably find winning tickets that generalize better, exceeding the test accuracy of the original network while matching its training accuracy. Test accuracy increases and then decreases as we prune, forming an *Occam's Hill* (Rasmussen & Ghahramani, 2001) where the original, overparameterized model has too much complexity (perhaps overfitting) and the extremely pruned model has too little. The conventional view of the relationship between compression and generalization is that compact hypotheses can better generalize (Rissanen, 1986). Recent theoretical work shows a similar link for neural networks, proving tighter generalization bounds for networks that can be compressed further (Zhou et al. (2018) for pruning/quantization and Arora et al. (2018) for noise robustness). The lottery ticket hypothesis offers a complementary perspective on this relationship—that larger networks might explicitly contain simpler representations.

**Implications for neural network optimization.** Winning tickets can reach accuracy equivalent to that of the original, unpruned network, but with significantly fewer parameters. This observation connects to recent work on the role of overparameterization in neural network training. For example, Du et al. (2019) prove that sufficiently overparameterized two-layer relu networks (with fixed-size second layers) trained with SGD converge to global optima. A key question, then, is whether the presence of a winning ticket is necessary or sufficient for SGD to optimize a neural network to a particular test accuracy. We conjecture (but do not empirically show) that SGD seeks out and trains a well-initialized subnetwork. By this logic, overparameterized networks are easier to train because they have more combinations of subnetworks that are potential winning tickets.

## 6    LIMITATIONS AND FUTURE WORK

We only consider vision-centric classification tasks on smaller datasets (MNIST, CIFAR10). We do not investigate larger datasets (namely Imagenet (Russakovsky et al., 2015)): iterative pruning is computationally intensive, requiring training a network 15 or more times consecutively for multiple trials. In future work, we intend to explore more efficient methods for finding winning tickets that will make it possible to study the lottery ticket hypothesis in more resource-intensive settings.

Sparse pruning is our only method for finding winning tickets. Although we reduce parameter-counts, the resulting architectures are not optimized for modern libraries or hardware. In future work, we intend to study other pruning methods from the extensive contemporary literature, such as structured pruning (which would produce networks optimized for contemporary hardware) and non-magnitude pruning methods (which could produce smaller winning tickets or find them earlier).

The winning tickets we find have initializations that allow them to match the performance of the unpruned networks at sizes too small for randomly-initialized networks to do the same. In future work, we intend to study the properties of these initializations that, in concert with the inductive biases of the pruned network architectures, make these networks particularly adept at learning.

On deeper networks (Resnet-18 and VGG-19), iterative pruning is unable to find winning tickets unless we train the networks with learning rate warmup. In future work, we plan to explore why warmup is necessary and whether other improvements to our scheme for identifying winning tickets could obviate the need for these hyperparameter modifications.

## 7 RELATED WORK

In practice, neural networks tend to be dramatically overparameterized. Distillation (Ba & Caruana, 2014; Hinton et al., 2015) and pruning (LeCun et al., 1990; Han et al., 2015) rely on the fact that parameters can be reduced while preserving accuracy. Even with sufficient capacity to memorize training data, networks naturally learn simpler functions (Zhang et al., 2016; Neyshabur et al., 2014; Arpit et al., 2017). Contemporary experience (Bengio et al., 2006; Hinton et al., 2015; Zhang et al., 2016) and Figure 1 suggest that overparameterized networks are easier to train. We show that dense networks contain sparse subnetworks capable of learning on their own starting from their original initializations. Several other research directions aim to train small or sparse networks.

**Prior to training.** Squeezenet (Iandola et al., 2016) and MobileNets (Howard et al., 2017) are specifically engineered image-recognition networks that are an order of magnitude smaller than standard architectures. Denil et al. (2013) represent weight matrices as products of lower-rank factors. Li et al. (2018) restrict optimization to a small, randomly-sampled subspace of the parameter space (meaning all parameters can still be updated); they successfully train networks under this restriction. We show that one need not even update all parameters to optimize a network, and we find winning tickets through a principled search process involving pruning. Our contribution to this class of approaches is to demonstrate that sparse, trainable networks exist within larger networks.

**After training.** Distillation (Ba & Caruana, 2014; Hinton et al., 2015) trains small networks to mimic the behavior of large networks; small networks are easier to train in this paradigm. Recent pruning work compresses large models to run with limited resources (e.g., on mobile devices). Although pruning is central to our experiments, we study why training needs the overparameterized networks that make pruning possible. LeCun et al. (1990) and Hassibi & Stork (1993) first explored pruning based on second derivatives. More recently, Han et al. (2015) showed per-weight magnitude-based pruning substantially reduces the size of image-recognition networks. Guo et al. (2016) restore pruned connections as they become relevant again. Han et al. (2017) and Jin et al. (2016) restore pruned connections to increase network capacity after small weights have been pruned and surviving weights fine-tuned. Other proposed pruning heuristics include pruning based on activations (Hu et al., 2016), redundancy (Mariet & Sra, 2016; Srinivas & Babu, 2015a), per-layer second derivatives (Dong et al., 2017), and energy/computation efficiency (Yang et al., 2017) (e.g., pruning convolutional filters (Li et al., 2016; Molchanov et al., 2016; Luo et al., 2017) or channels (He et al., 2017)). Cohen et al. (2016) observe that convolutional filters are sensitive to initialization ("The Filter Lottery"); throughout training, they randomly reinitialize unimportant filters.

**During training.** Bellec et al. (2018) train with sparse networks and replace weights that reach zero with new random connections. Srinivas et al. (2017) and Louizos et al. (2018) learn gating variables that minimize the number of nonzero parameters. Narang et al. (2017) integrate magnitude-based pruning into training. Gal & Ghahramani (2016) show that dropout approximates Bayesian inference in Gaussian processes. Bayesian perspectives on dropout learn dropout probabilities during training (Gal et al., 2017; Kingma et al., 2015; Srinivas & Babu, 2016). Techniques that learn per-weight, per-unit (Srinivas & Babu, 2016), or structured dropout probabilities naturally (Molchanov et al., 2017; Neklyudov et al., 2017) or explicitly (Louizos et al., 2017; Srinivas & Babu, 2015b) prune and sparsify networks during training as dropout probabilities for some weights reach 1. In contrast, we train networks at least once to find winning tickets. These techniques might also find winning tickets, or, by inducing sparsity, might beneficially interact with our methods.

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

## A  ACKNOWLEDGMENTS

We gratefully acknowledge IBM, which—through the MIT-IBM Watson AI Lab—contributed the computational resources necessary to conduct the experiments in this paper. We particularly thank IBM researchers German Goldszmidt, David Cox, Ian Molloy, and Benjamin Edwards for their generous contributions of infrastructure, technical support, and feedback. We also wish to thank Aleksander Madry, Shafi Goldwasser, Ed Felten, David Bieber, Karolina Dziugaite, Daniel Weitzner, and R. David Edelman for support, feedback, and helpful discussions over the course of this project. This work was support in part by the Office of Naval Research (ONR N00014-17-1-2699).

## B  ITERATIVE PRUNING STRATEGIES

In this Appendix, we examine two different ways of structuring the iterative pruning strategy that we use throughout the main body of the paper to find winning tickets.

**Strategy 1: Iterative pruning with resetting.**

1. Randomly initialize a neural network $f(x; m \odot \theta)$ where $\theta = \theta_0$ and $m = 1^{|\theta|}$ is a mask.
2. Train the network for $j$ iterations, reaching parameters $m \odot \theta_j$.
3. Prune $s\%$ of the parameters, creating an updated mask $m'$ where $P_{m'} = (P_m - s)\%$.
4. Reset the weights of the remaining portion of the network to their values in $\theta_0$. That is, let $\theta = \theta_0$.
5. Let $m = m'$ and repeat steps 2 through 4 until a sufficiently pruned network has been obtained.

**Strategy 2: Iterative pruning with continued training.**

1. Randomly initialize a neural network $f(x; m \odot \theta)$ where $\theta = \theta_0$ and $m = 1^{|\theta|}$ is a mask.
2. Train the network for $j$ iterations.
3. Prune $s\%$ of the parameters, creating an updated mask $m'$ where $P_{m'} = (P_m - s)\%$.
4. Let $m = m'$ and repeat steps 2 and 3 until a sufficiently pruned network has been obtained.
5. Reset the weights of the remaining portion of the network to their values in $\theta_0$. That is, let $\theta = \theta_0$.

The difference between these two strategies is that, after each round of pruning, Strategy 2 retrains using the already-trained weights, whereas Strategy 1 resets the network weights back to their initial values before retraining. In both cases, after the network has been sufficiently pruned, its weights are reset back to the original initializations.

Figures 9 and 10 compare the two strategies on the Lenet and Conv-2/4/6 architectures on the hyperparameters we select in Appendices G and H. In all cases, the Strategy 1 maintains higher validation accuracy and faster early-stopping times to smaller network sizes.

## C  EARLY STOPPING CRITERION

Throughout this paper, we are interested in measuring the speed at which networks learn. As a proxy for this quantity, we measure the iteration at which an early-stopping criterion would end training. The specific criterion we employ is the iteration of minimum validation loss. In this Subsection, we further explain that criterion.

Validation and test loss follow a pattern where they decrease early in the training process, reach a minimum, and then begin to increase as the model overfits to the training data. Figure 11 shows an example of the validation loss as training progresses; these graphs use Lenet, iterative pruning, and Adam with a learning rate of 0.0012 (the learning rate we will select in the following subsection). This Figure shows the validation loss corresponding to the test accuracies in Figure 3.

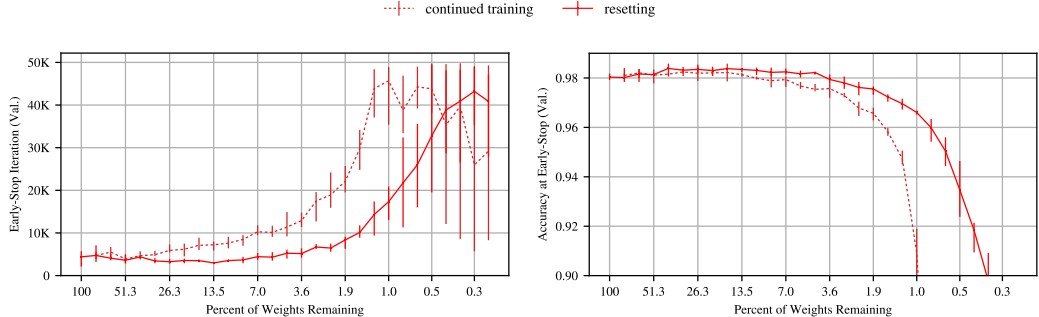

Figure 9: The early-stopping iteration and accuracy at early-stopping of the iterative lottery ticket experiment on the Lenet architecture when iteratively pruned using the resetting and continued training strategies.

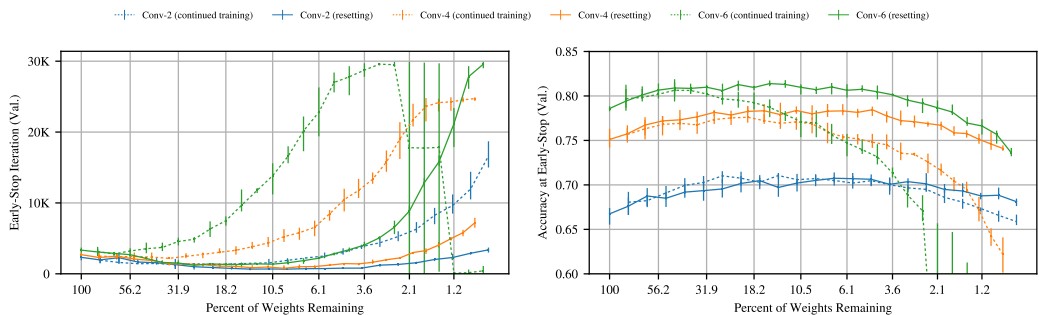

Figure 10: The early-stopping iteration and accuracy at early-stopping of the iterative lottery ticket experiment on the Conv-2, Conv-4, and Conv-6 architectures when iteratively pruned using the resetting and continued training strategies.

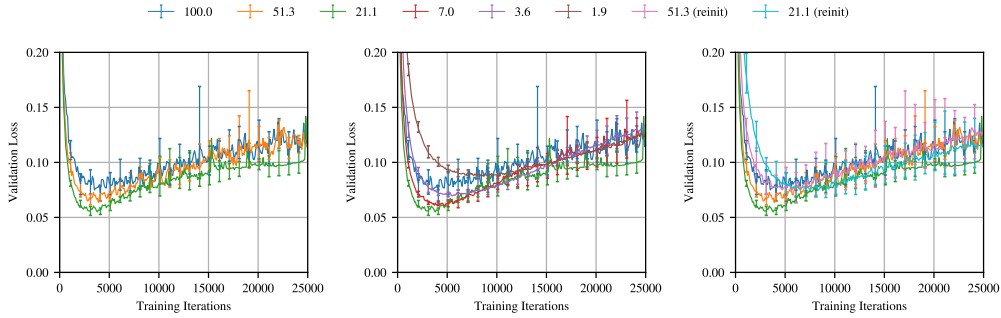

Figure 11: The validation loss data corresponding to Figure 3, i.e., the validation loss as training progresses for several different levels of pruning in the iterative pruning experiment. Each line is the average of five training runs at the same level of iterative pruning; the labels are the percentage of weights from the original network that remain after pruning. Each network was trained with Adam at a learning rate of 0.0012. The left graph shows winning tickets that learn increasingly faster than the original network and reach lower loss. The middle graph shows winning tickets that learn increasingly slower after the fastest early-stopping time has been reached. The right graph contrasts the loss of winning tickets to the loss of randomly reinitialized networks.

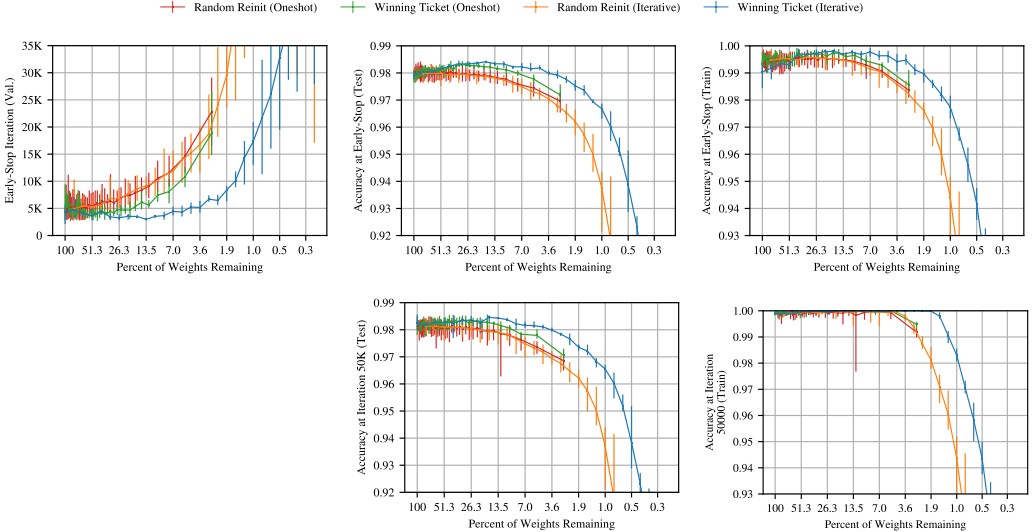

Figure 12: Figure 4 augmented with a graph of the training accuracy at the end of 50,000 iterations.

In all cases, validation loss initially drops, after which it forms a clear bottom and then begins increasing again. Our early-stopping criterion identifies this bottom. We consider networks that reach this moment sooner to have learned "faster." In support of this notion, the ordering in which each experiment meets our early-stopping criterion in Figure 3 is the same order in which each experiment reaches a particular test accuracy threshold in Figure 3.

Throughout this paper, in order to contextualize this learning speed, we also present the test accuracy of the network at the iteration of minimum validation loss. In the main body of the paper, we find that winning tickets both arrive at early-stopping sooner and reach higher test accuracy at this point.

## D    TRAINING ACCURACY FOR LOTTERY TICKET EXPERIMENTS

This Appendix accompanies Figure 4 (the accuracy and early-stopping iterations of Lenet on MNIST from Section 2) and Figure 5 (the accuracy and early-stopping iterations of Conv-2, Conv-4, and Conv-6 in Section Section 3) in the main body of the paper. Those figures show the iteration of early-stopping, the test accuracy at early-stopping, the training accuracy at early-stopping, and the test accuracy at the end of the training process. However, we did not have space to include a graph of the training accuracy at the end of the training process, which we assert in the main body of the paper to be 100% for all but the most heavily pruned networks. In this Appendix, we include those additional graphs in Figure 12 (corresponding to Figure 4) and Figure 13 (corresponding to Figure 5). As we describe in the main body of the paper, training accuracy reaches 100% in all cases for all but the most heavily pruned networks. However, training accuracy remains at 100% longer for winning tickets than for randomly reinitialized networks.

## E    COMPARING RANDOM REINITIALIZATION AND RANDOM SPARSITY

In this Appendix, we aim to understand the relative performance of randomly reinitialized winning tickets and randomly sparse networks.

1. Networks found via iterative pruning with the original initializations (blue in Figure 14).

2. Networks found via iterative pruning that are randomly reinitialized (orange in Figure 14).

3. Random sparse subnetworks with the same number of parameters as those found via iterative pruning (green in Figure 14).

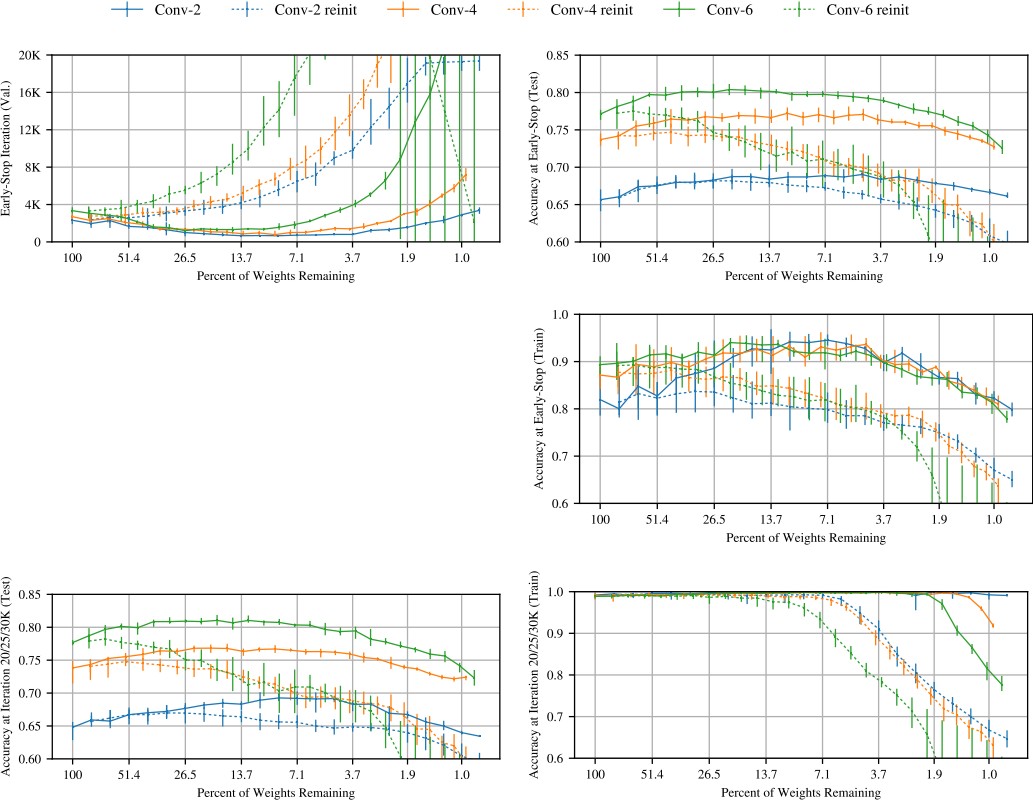

Figure 13: Figure 5 augmented with a graph of the training accuracy at the end of the training process.

Figure 14 shows this comparison for all of the major experiments in this paper. For the fully-connected Lenet architecture for MNIST, we find that the randomly reinitialized networks outperform random sparsity. However, for all of the other, convolutional networks studied in this paper, there is no significant difference in performance between the two. We hypothesize that the fully-connected network for MNIST sees these benefits because only certain parts of the MNIST images contain useful information for classification, meaning connections in some parts of the network will be more valuable than others. This is less true with convolutions, which are not constrained to any one part of the input image.

## F    EXAMINING WINNING TICKETS

In this Appendix, we examine the structure of winning tickets to gain insight into why winning tickets are able to learn effectively even when so heavily pruned. Throughout this Appendix, we study the winning tickets from the Lenet architecture trained on MNIST. Unless otherwise stated, we use the same hyperparameters as in Section 2: glorot initialization and adam optimization.

### F.1    WINNING TICKET INITIALIZATION (ADAM)

Figure 15 shows the distributions of winning ticket initializations for four different levels of $P_m$. To clarify, these are the distributions of the initial weights of the connections that have survived the pruning process. The blue, orange, and green lines show the distribution of weights for the first hidden layer, second hidden layer, and output layer, respectively. The weights are collected from five different trials of the lottery ticket experiment, but the distributions for each individual trial closely mirror those aggregated from across all of the trials. The histograms have been normalized so that the area under each curve is 1.

The left-most graph in Figure 15 shows the initialization distributions for the unpruned networks. We use glorot initialization, so each of the layers has a different standard deviation. As the network is pruned, the first hidden layer maintains its distribution. However, the second hidden layer and the output layer become increasingly bimodal, with peaks on either side of 0. Interestingly, the peaks are asymmetric: the second hidden layer has more positive initializations remaining than negative initializations, and the reverse is true for the output layer.

The connections in the second hidden layer and output layer that survive the pruning process tend to have higher magnitude-initializations. Since we find winning tickets by pruning the connections with the lowest magnitudes in each layer at the *end*, the connections with the lowest-magnitude initializations must still have the lowest-magnitude weights at the end of training. A different trend holds for the input layer: it maintains its distribution, meaning a connection's initialization has less relation to its final weight.

### F.2    WINNING TICKET INITIALIZATIONS (SGD)

We also consider the winning tickets obtained when training the network with SGD learning rate 0.8 (selected as described in Appendix G). The bimodal distributions from Figure 15 are present across all layers (see Figure 16. The connections with the highest-magnitude initializations are more likely to survive the pruning process, meaning winning ticket initializations have a bimodal distribution with peaks on opposite sides of 0. Just as with the adam-optimized winning tickets, these peaks are of different sizes, with the first hidden layer favoring negative initializations and the second hidden layer and output layer favoring positive initializations. Just as with the adam results, we confirm that each individual trial evidences the same asymmetry as the aggregate graphs in Figure 16.

### F.3    REINITIALIZING FROM WINNING TICKET INITIALIZATIONS

Considering that the initialization distributions of winning tickets $\mathcal{D}_m$ are so different from the Gaussian distribution $\mathcal{D}$ used to initialize the unpruned network, it is natural to ask whether randomly reinitializing winning tickets from $\mathcal{D}_m$ rather than $\mathcal{D}$ will improve winning ticket performance. We do not find this to be the case. Figure 17 shows the performance of winning tickets whose initializations are randomly sampled from the distribution of initializations contained in the winning tickets for

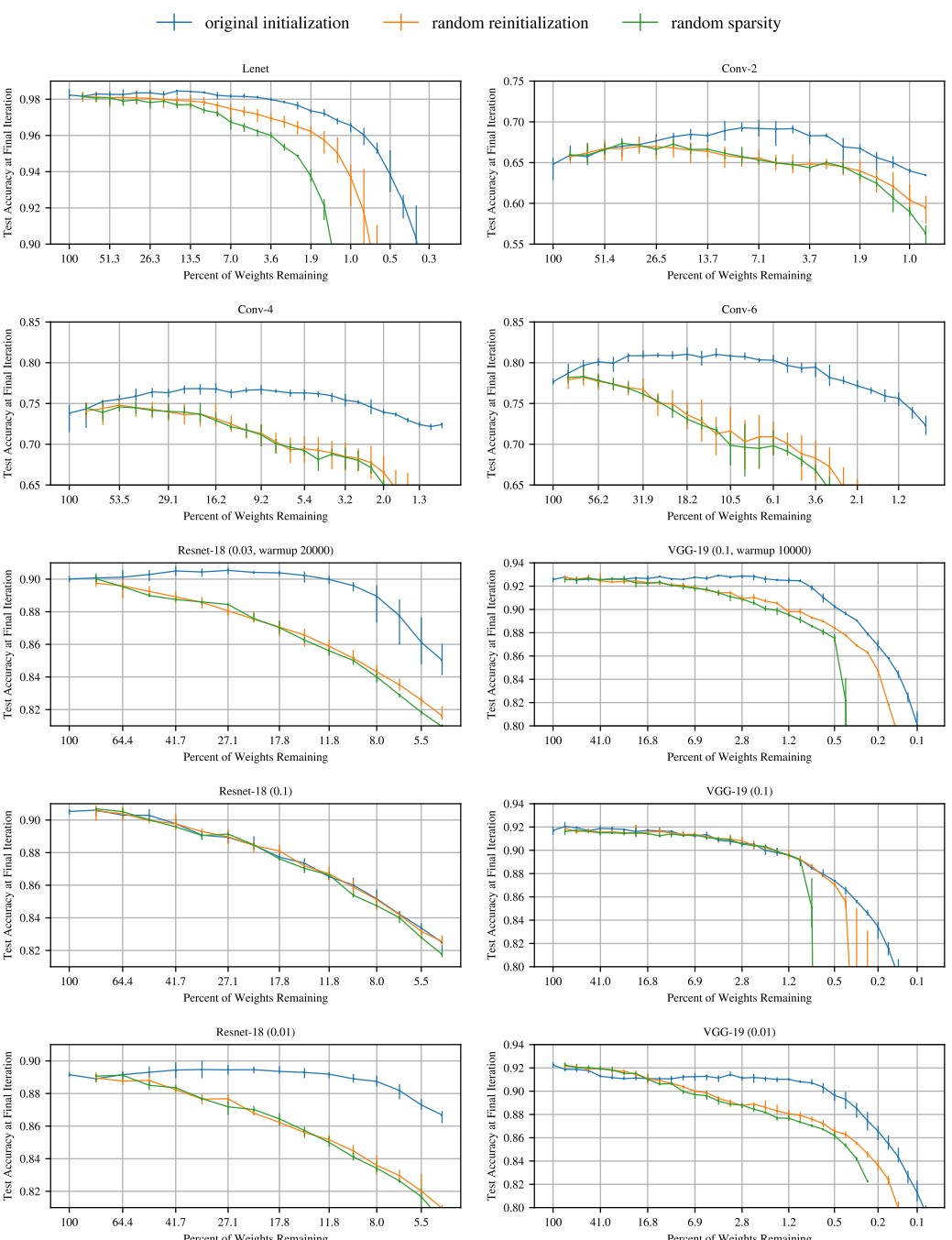

Figure 14: The test accuracy at the final iteration for each of the networks studied in this paper.

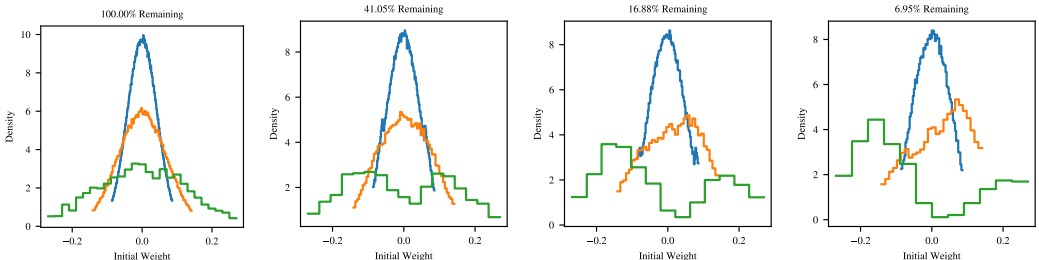

Figure 15: The distribution of initializations in winning tickets pruned to the levels specified in the titles of each plot. The blue, orange, and green lines show the distributions for the first hidden layer, second hidden layer, and output layer of the Lenet architecture for MNIST when trained with the adam optimizer and the hyperparameters used in 2. The distributions have been normalized so that the area under each curve is 1.

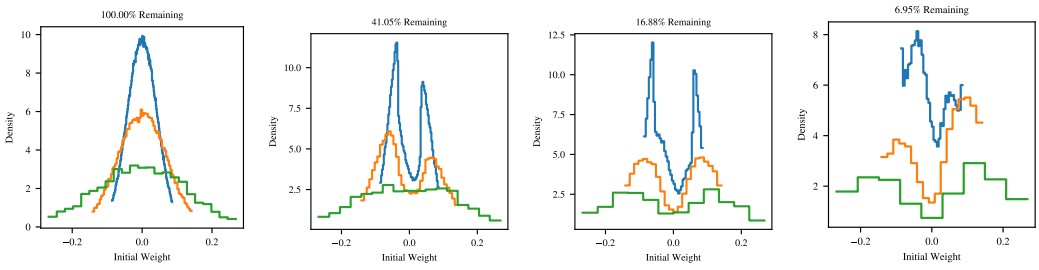

Figure 16: Same as Figure 15 where the network is trained with SGD at rate 0.8.

adam. More concretely, let $\mathcal{D}_m = \{\theta_0^{(i)} | m^{(i)} = 1\}$ be the set of initializations found in the winning ticket with mask $m$. We sample a new set of parameters $\theta_0' \sim \mathcal{D}_m$ and train the network $f(x; m \odot \theta_0')$. We perform this sampling on a per-layer basis. The results of this experiment are in Figure 17. Winning tickets reinitialized from $\mathcal{D}_m$ perform little better than when randomly reinitialized from $\mathcal{D}$. We attempted the same experiment with the SGD-trained winning tickets and found similar results.

### F.4 PRUNING AT ITERATION 0

One other way of interpreting the graphs of winning ticket initialization distributions is as follows: weights that begin small stay small, get pruned, and never become part of the winning ticket. (The only exception to this characterization is the first hidden layer for the adam-trained winning tickets.) If this is the case, then perhaps low-magnitude weights were never important to the network and can be pruned from the very beginning. Figure 18 shows the result of attempting this pruning strategy. Winning tickets selected in this fashion perform even worse than when they are found by iterative

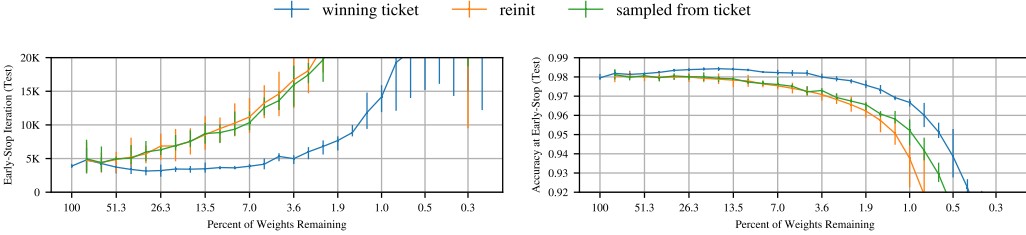

Figure 17: The performance of the winning tickets of the Lenet architecture for MNIST when the layers are randomly reinitialized from the distribution of initializations contained in the winning ticket of the corresponding size.

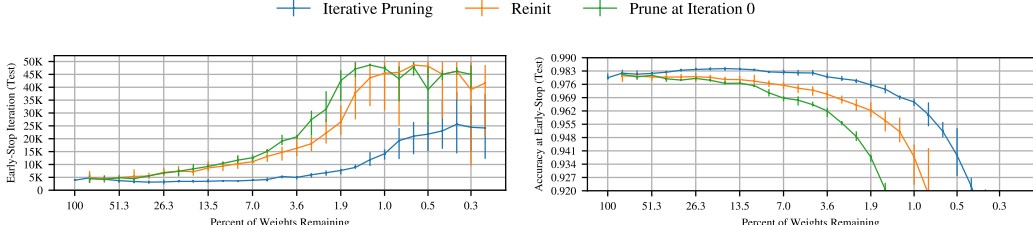

Figure 18: The performance of the winning tickets of the Lenet architecture for MNIST when magnitude pruning is performed before the network is ever trained. The network is subsequently trained with adam.

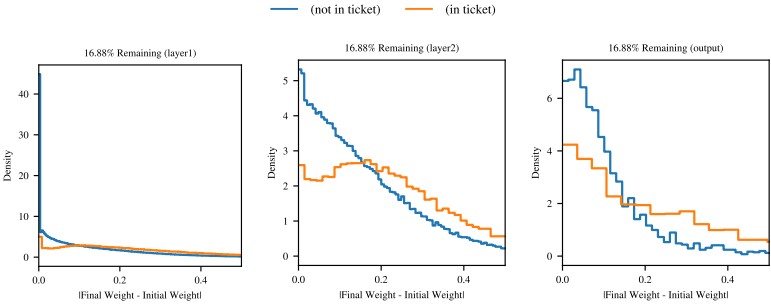

Figure 19: Between the first and last training iteration of the unpruned network, the magnitude by which weights in the network change. The blue line shows the distribution of magnitudes for weights that are not in the eventual winning ticket; the orange line shows the distribution of magnitudes for weights that are in the eventual winning ticket.

pruning and randomly reinitialized. We attempted the same experiment with the SGD-trained winning tickets and found similar results.

### F.5 COMPARING INITIAL AND FINAL WEIGHTS IN WINNING TICKETS

In this subsection, we consider winning tickets in the context of the larger optimization process. To do so, we examine the initial and final weights of the unpruned network from which a winning ticket derives to determine whether weights that will eventually comprise a winning ticket exhibit properties that distinguish them from the rest of the network.

We consider the magnitude of the difference between initial and final weights. One possible rationale for the success of winning tickets is that they already happen to be close to the optimum that gradient descent eventually finds, meaning that winning ticket weights should change by a smaller amount than the rest of the network. Another possible rationale is that winning tickets are well placed in the optimization landscape for gradient descent to optimize productively, meaning that winning ticket weights should change by a larger amount than the rest of the network. Figure 19 shows that winning ticket weights tend to change by a larger amount then weights in the rest of the network, evidence that does not support the rationale that winning tickets are already close to the optimum.

It is notable that such a distinction exists between the two distributions. One possible explanation for this distinction is that the notion of a winning ticket may indeed be a natural part of neural network optimization. Another is that magnitude-pruning biases the winning tickets we find toward those containing weights that change in the direction of higher magnitude. Regardless, it offers hope that winning tickets may be discernible earlier in the training process (or after a single training run), meaning that there may be more efficient methods for finding winning tickets than iterative pruning.

Figure 20 shows the directions of these changes. It plots the difference between the magnitude of the final weight and the magnitude of the initial weight, i.e., whether the weight moved toward or away

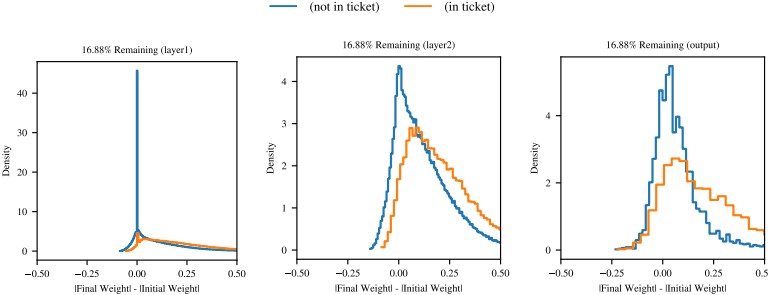

Figure 20: Between the first and last training iteration of the unpruned network, the magnitude by which weights move away from 0. The blue line shows the distribution of magnitudes for weights that are not in the eventual winning ticket; the orange line shows the distribution of magnitudes for weights that are in the eventual winning ticket.

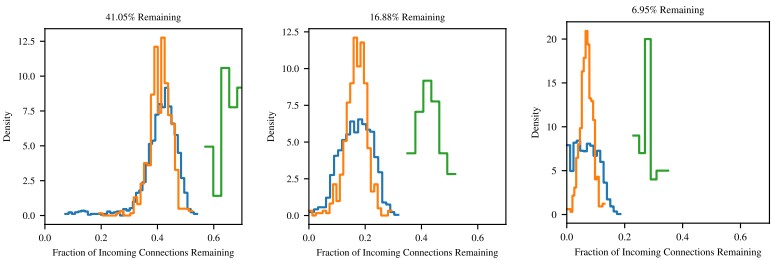

Figure 21: The fraction of incoming connections that survive the pruning process for each node in each layer of the Lenet architecture for MNIST as trained with adam.

from 0. In general, winning ticket weights are more likely to increase in magnitude (that is, move away from 0) than are weights that do not participate in the eventual winning ticket.

## F.6 WINNING TICKET CONNECTIVITY

In this Subsection, we study the connectivity of winning tickets. Do some hidden units retain a large number of incoming connections while others fade away, or does the network retain relatively even sparsity among all units as it is pruned? We find the latter to be the case when examining the incoming connectivity of network units: for both adam and SGD, each unit retains a number of incoming connections approximately in proportion to the amount by which the overall layer has been pruned. Figures 21 and 22 show the fraction of incoming connections that survive the pruning process for each node in each layer. Recall that we prune the output layer at half the rate as the rest of the network, which explains why it has more connectivity than the other layers of the network.

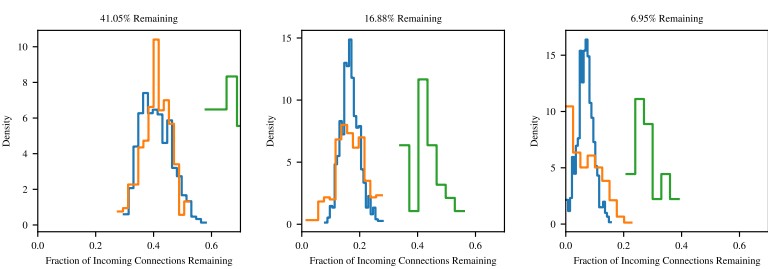

Figure 22: Same as Figure 21 where the network is trained with SGD at rate 0.8.

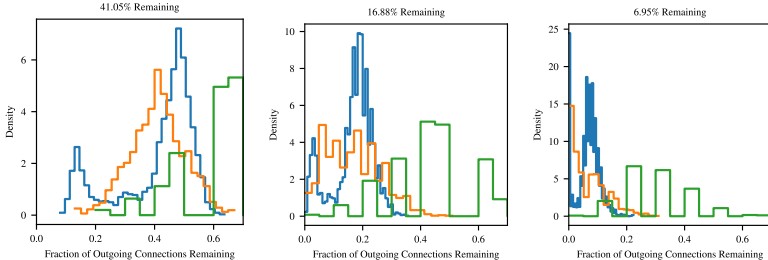

Figure 23: The fraction of outgoing connections that survive the pruning process for each node in each layer of the Lenet architecture for MNIST as trained with adam. The blue, orange, and green lines are the outgoing connections from the input layer, first hidden layer, and second hidden layer, respectively.

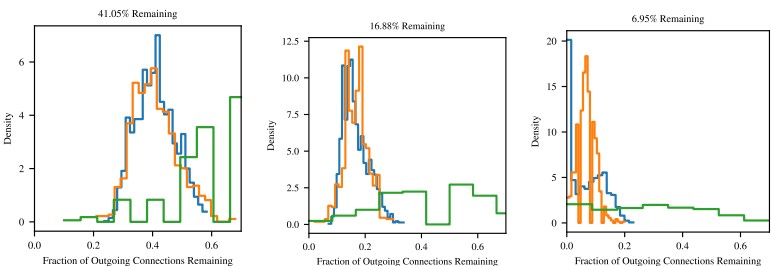

Figure 24: Same as Figure 23 where the network is trained with SGD at rate 0.8.

However, this is not the case for the outgoing connections. To the contrary, for the adam-trained networks, certain units retain far more outgoing connections than others (Figure 23). The distributions are far less smooth than those for the incoming connections, suggesting that certain features are far more useful to the network than others. This is not unexpected for a fully-connected network on a task like MNIST, particularly for the input layer: MNIST images contain centered digits, so the pixels around the edges are not likely to be informative for the network. Indeed, the input layer has two peaks, one larger peak for input units with a high number of outgoing connections and one smaller peak for input units with a low number of outgoing connections. Interestingly, the adam-trained winning tickets develop a much more uneven distribution of outgoing connectivity for the input layer than does the SGD-trained network (Figure 24).

## F.7 ADDING NOISE TO WINNING TICKETS

In this Subsection, we explore the extent to which winning tickets are robust to Gaussian noise added to their initializations. In the main body of the paper, we find that randomly reinitializing a winning ticket substantially slows its learning and reduces its eventual test accuracy. In this Subsection, we study a less extreme way of perturbing a winning ticket. Figure 25 shows the effect of adding Gaussian noise to the winning ticket initializations. The standard deviation of the noise distribution of each layer is a multiple of the standard deviation of the layer's initialization Figure 25 shows noise distributions with standard deviation $0.5\sigma$, $\sigma$, $2\sigma$, and $3\sigma$. Adding Gaussian noise reduces the test accuracy of a winning ticket and slows its ability to learn, again demonstrating the importance of the original initialization. As more noise is added, accuracy decreases. However, winning tickets are surprisingly robust to noise. Adding noise of $0.5\sigma$ barely changes winning ticket accuracy. Even after adding noise of $3\sigma$, the winning tickets continue to outperform the random reinitialization experiment.

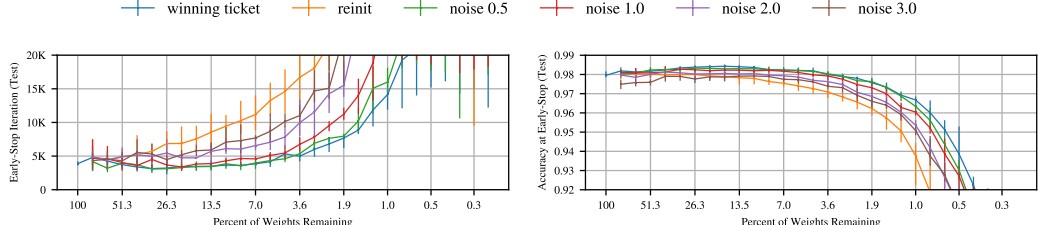

Figure 25: The performance of the winning tickets of the Lenet architecture for MNIST when Gaussian noise is added to the initializations. The standard deviations of the noise distributions for each layer are a multiple of the standard deviations of the initialization distributions; in this Figure, we consider multiples 0.5, 1, 2, and 3.

# G    HYPERPARAMETER EXPLORATION FOR FULLY-CONNECTED NETWORKS

This Appendix accompanies Section 2 of the main paper. It explores the space of hyperparameters for the Lenet architecture evaluated in Section 2 with two purposes in mind:

1. To explain the hyperparameters selected in the main body of the paper.
2. To evaluate the extent to which the lottery ticket experiment patterns extend to other choices of hyperparameters.

## G.1    EXPERIMENTAL METHODOLOGY

This Section considers the fully-connected Lenet architecture (LeCun et al., 1998), which comprises two fully-connected hidden layers and a ten unit output layer, on the MNIST dataset. Unless otherwise stated, the hidden layers have 300 and 100 units each.

The MNIST dataset consists of 60,000 training examples and 10,000 test examples. We randomly sampled a 5,000-example validation set from the training set and used the remaining 55,000 training examples as our training set for the rest of the paper (including Section 2). The hyperparameter selection experiments throughout this Appendix are evaluated using the validation set for determining both the iteration of early-stopping and the accuracy at early-stopping; the networks in the main body of this paper (which make use of these hyperparameters) have their accuracy evaluated on the test set. The training set is presented to the network in mini-batches of 60 examples; at each epoch, the entire training set is shuffled.

Unless otherwise noted, each line in each graph comprises data from three separate experiments. The line itself traces the average performance of the experiments and the error bars indicate the minimum and maximum performance of any one experiment.

Throughout this Appendix, we perform the lottery ticket experiment iteratively with a pruning rate of 20% per iteration (10% for the output layer); we justify the choice of this pruning rate later in this Appendix. Each layer of the network is pruned independently. On each iteration of the lottery ticket experiment, the network is trained for 50,000 training iterations regardless of when early-stopping occurs; in other words, no validation or test data is taken into account during the training process, and early-stopping times are determined retroactively by examining validation performance. We evaluate validation and test performance every 100 iterations.

For the main body of the paper, we opt to use the Adam optimizer (Kingma & Ba, 2014) and Gaussian Glorot initialization (Glorot & Bengio, 2010). Although we can achieve more impressive results on the lottery ticket experiment with other hyperparameters, we intend these choices to be as generic as possible in an effort to minimize the extent to which our main results depend on hand-chosen hyperparameters. In this Appendix, we select the learning rate for Adam that we use in the main body of the paper.

In addition, we consider a wide range of other hyperparameters, including other optimization algorithms (SGD with and without momentum), initialization strategies (Gaussian distributions

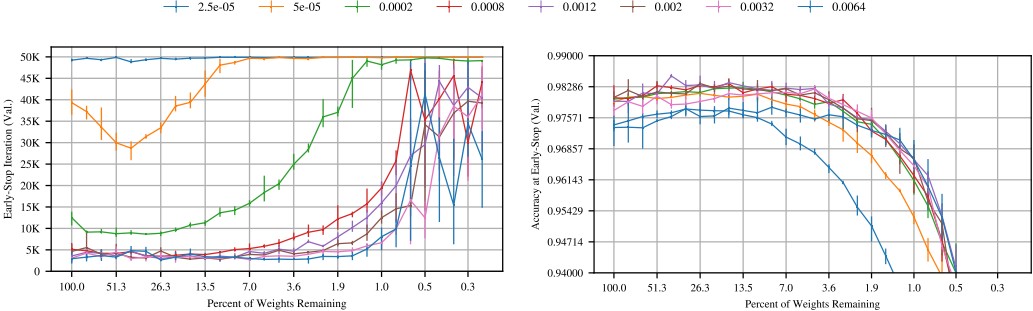

Figure 26: The early-stopping iteration and validation accuracy at that iteration of the iterative lottery ticket experiment on the Lenet architecture trained with MNIST using the Adam optimizer at various learning rates. Each line represents a different learning rate.

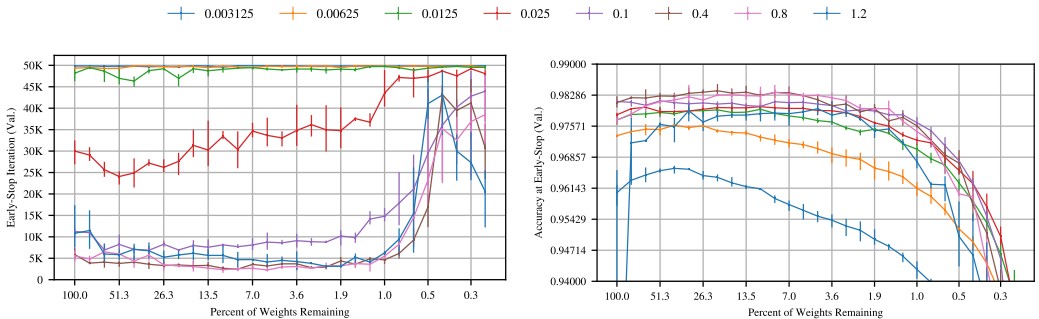

Figure 27: The early-stopping iteration and validation accuracy at that iteration of the iterative lottery ticket experiment on the Lenet architecture trained with MNIST using stochastic gradient descent at various learning rates.

with various standard deviations), network sizes (larger and smaller hidden layers), and pruning strategies (faster and slower pruning rates). In each experiment, we vary the chosen hyperparameter while keeping all others at their default values (Adam with the chosen learning rate, Gaussian Glorot initialization, hidden layers with 300 and 100 units). The data presented in this appendix was collected by training variations of the Lenet architecture more than 3,000 times.

## G.2 LEARNING RATE

In this Subsection, we perform the lottery ticket experiment on the Lenet architecture as optimized with Adam, SGD, and SGD with momentum at various learning rates.

Here, we select the learning rate that we use for Adam in the main body of the paper. Our criteria for selecting the learning rate are as follows:

1. On the unpruned network, it should minimize training iterations necessary to reach early-stopping and maximize validation accuracy at that iteration. That is, it should be a reasonable hyperparameter for optimizing the unpruned network even if we are not running the lottery ticket experiment.

2. When running the iterative lottery ticket experiment, it should make it possible to match the early-stopping iteration and accuracy of the original network with as few parameters as possible.

3. Of those options that meet (1) and (2), it should be on the conservative (slow) side so that it is more likely to productively optimize heavily pruned networks under a variety of conditions with a variety of hyperparameters.

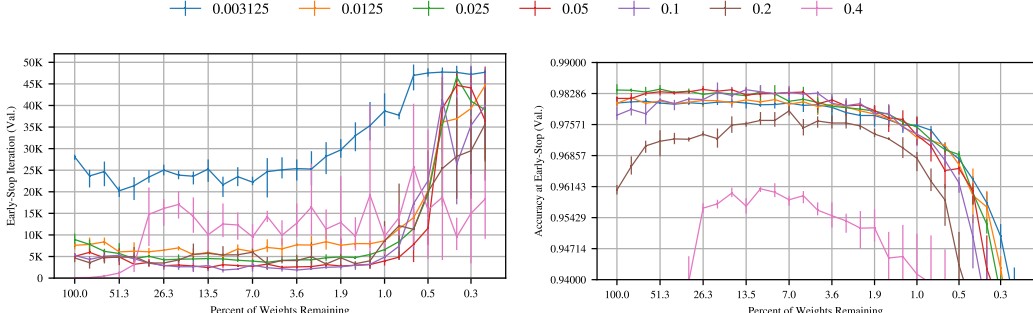

Figure 28: The early-stopping iteration and validation accuracy at that iteration of the iterative lottery ticket experiment on the Lenet architecture trained with MNIST using stochastic gradient descent with momentum (0.9) at various learning rates.

Figure 26 shows the early-stopping iteration and validation accuracy at that iteration of performing the iterative lottery ticket experiment with the Lenet architecture optimized with Adam at various learning rates. According to the graph on the right of Figure 26, several learning rates between 0.0002 and 0.002 achieve similar levels of validation accuracy on the original network and maintain that performance to similar levels as the network is pruned. Of those learning rates, 0.0012 and 0.002 produce the fastest early-stopping times and maintain them to the smallest network sizes. We choose 0.0012 due to its higher validation accuracy on the unpruned network and in consideration of criterion (3) above.

We note that, across all of these learning rates, the lottery ticket pattern (in which learning becomes faster and validation accuracy increases with iterative pruning) remains present. Even for those learning rates that did not satisfy the early-stopping criterion within 50,000 iterations (2.5e-05 and 0.0064) still showed accuracy improvements with pruning.

### G.3 OTHER OPTIMIZATION ALGORITHMS

### G.3.1 SGD

Here, we explore the behavior of the lottery ticket experiment when the network is optimized with stochastic gradient descent (SGD) at various learning rates. The results of doing so appear in Figure 27. The lottery ticket pattern appears across all learning rates, including those that fail to satisfy the early-stopping criterion within 50,000 iterations. SGD learning rates 0.4 and 0.8 reach early-stopping in a similar number of iterations as the best Adam learning rates (0.0012 and 0.002) but maintain this performance when the network has been pruned further (to less than 1% of its original size for SGD vs. about 3.6% of the original size for Adam). Likewise, on pruned networks, these SGD learning rates achieve equivalent accuracy to the best Adam learning rates, and they maintain that high accuracy when the network is pruned as much as the Adam learning rates.

### G.3.2 MOMENTUM

Here, we explore the behavior of the lottery ticket experiment when the network is optimized with SGD with momentum (0.9) at various learning rates. The results of doing so appear in Figure 28. Once again, the lottery ticket pattern appears across all learning rates, with learning rates between 0.025 and 0.1 maintaining high validation accuracy and faster learning for the longest number of pruning iterations. Learning rate 0.025 achieves the highest validation accuracy on the unpruned network; however, its validation accuracy never increases as it is pruned, instead decreasing gradually, and higher learning rates reach early-stopping faster.

### G.4 ITERATIVE PRUNING RATE

When running the iterative lottery ticket experiment on Lenet, we prune each layer of the network separately at a particular rate. That is, after training the network, we prune $k\%$ of the weights in

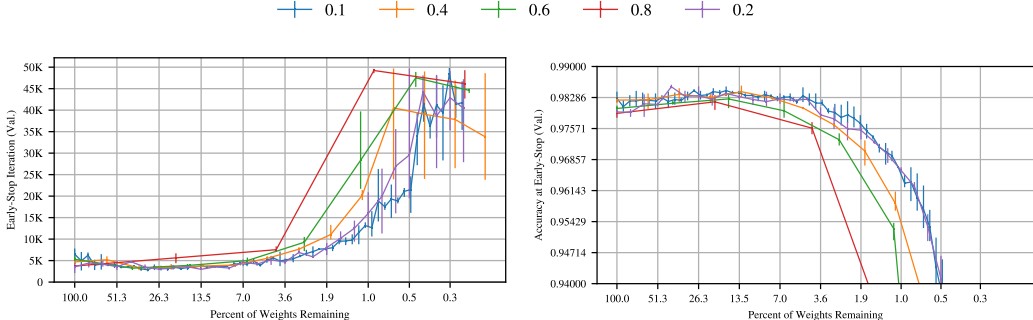

Figure 29: The early-stopping iteration and validation accuracy at that iteration of the iterative lottery ticket experiment when pruned at different rates. Each line represents a different *pruning rate*—the percentage of lowest-magnitude weights that are pruned from each layer after each training iteration.

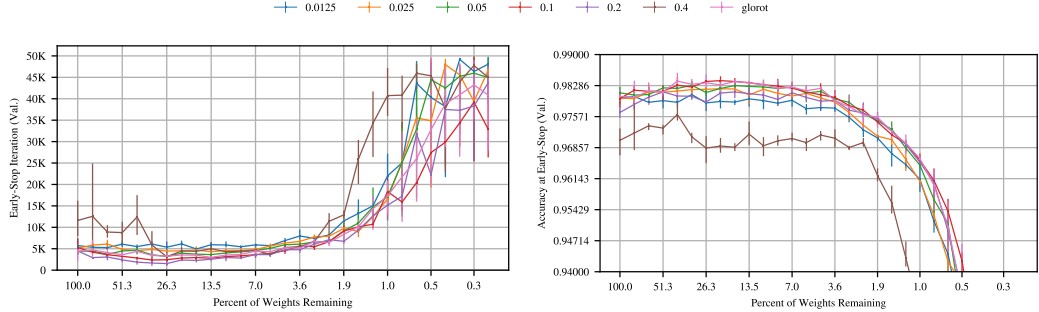

Figure 30: The early-stopping iteration and validation accuracy at that iteration of the iterative lottery ticket experiment initialized with Gaussian distributions with various standard deviations. Each line is a different standard deviation for a Gaussian distribution centered at 0.

each layer ($\frac{k}{2}\%$ of the weights in the output layer) before resetting the weights to their original initializations and training again. In the main body of the paper, we find that iterative pruning finds smaller winning tickets than one-shot pruning, indicating that pruning too much of the network at once diminishes performance. Here, we explore different values of $k$.

Figure 29 shows the effect of the amount of the network pruned on each pruning iteration on early-stopping time and validation accuracy. There is a tangible difference in learning speed and validation accuracy at early-stopping between the lowest pruning rates (0.1 and 0.2) and higher pruning rates (0.4 and above). The lowest pruning rates reach higher validation accuracy and maintain that validation accuracy to smaller network sizes; they also maintain fast early-stopping times to smaller network sizes. For the experiments throughout the main body of the paper and this Appendix, we use a pruning rate of 0.2, which maintains much of the accuracy and learning speed of 0.1 while reducing the number of training iterations necessary to get to smaller network sizes.

In all of the Lenet experiments, we prune the output layer at half the rate of the rest of the network. Since the output layer is so small (1,000 weights out of 266,000 for the overall Lenet architecture), we found that pruning it reaches a point of diminishing returns much earlier the other layers.

### G.5 INITIALIZATION DISTRIBUTION

To this point, we have considered only a Gaussian Glorot (Glorot & Bengio, 2010) initialization scheme for the network. Figure 30 performs the lottery ticket experiment while initializing the Lenet architecture from Gaussian distributions with a variety of standard deviations. The networks were optimized with Adam at the learning rate chosen earlier. The lottery ticket pattern continues to appear across all standard deviations. When initialized from a Gaussian distribution with standard deviation

0.1, the Lenet architecture maintained high validation accuracy and low early-stopping times for the longest, approximately matching the performance of the Glorot-initialized network.

## G.6 NETWORK SIZE

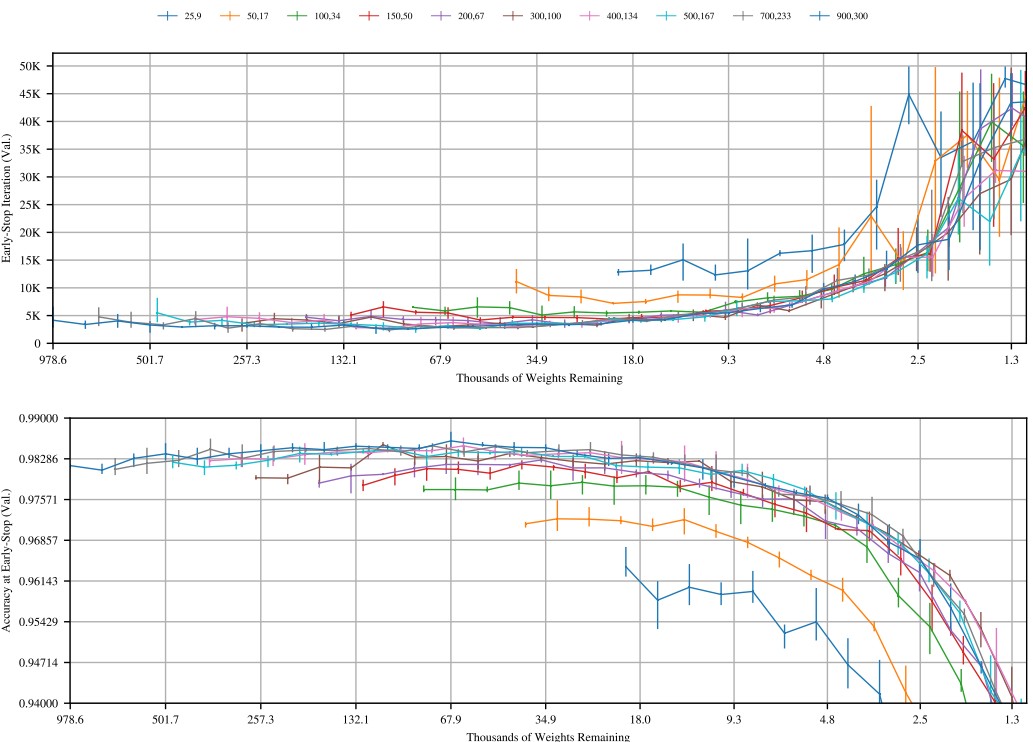

Figure 31: The early-stopping iteration and validation accuracy at at that iteration of the iterative lottery ticket experiment on the Lenet architecture with various layer sizes. The label for each line is the size of the first and second hidden layers of the network. All networks had Gaussian Glorot initialization and were optimized with Adam (learning rate 0.0012). Note that the x-axis of this plot charts the number of *weights* remaining, while all other graphs in this section have charted the *percent* of weights remaining.

Throughout this section, we have considered the Lenet architecture with 300 units in the first hidden layer and 100 units in the second hidden layer. Figure 31 shows the early-stopping iterations and validation accuracy at that iteration of the Lenet architecture with several other layer sizes. All networks we tested maintain the 3:1 ratio between units in the first hidden layer and units in the second hidden layer.

The lottery ticket hypothesis naturally invites a collection of questions related to network size. Generalizing, those questions tend to take the following form: according to the lottery ticket hypothesis, do larger networks, which contain more subnetworks, find "better" winning tickets? In line with the generality of this question, there are several different answers.

If we evaluate a winning ticket by the accuracy it achieves, then larger networks do find better winning tickets. The right graph in Figure 31 shows that, for any particular number of weights (that is, any particular point on the x-axis), winning tickets derived from initially larger networks reach higher accuracy. Put another way, in terms of accuracy, the lines are approximately arranged from bottom to top in increasing order of network size. It is possible that, since larger networks have more subnetworks, gradient descent found a better winning ticket. Alternatively, the initially larger networks have more units even when pruned to the same number of weights as smaller networks, meaning they are able to contain sparse subnetwork configurations that cannot be expressed by initially smaller networks.

If we evaluate a winning ticket by the time necessary for it to reach early-stopping, then larger networks have less of an advantage. The left graph in Figure 31 shows that, in general, early-stopping iterations do not vary greatly between networks of different initial sizes that have been pruned to the same number of weights. Upon exceedingly close inspection, winning tickets derived from initially larger networks tend to learn marginally faster than winning tickets derived from initially smaller networks, but these differences are slight.

If we evaluate a winning ticket by the size at which it returns to the same accuracy as the original network, the large networks do not have an advantage. Regardless of the initial network size, the right graph in Figure 31 shows that winning tickets return to the accuracy of the original network when they are pruned to between about 9,000 and 15,000 weights.

## H  HYPERPARAMETER EXPLORATION FOR CONVOLUTIONAL NETWORKS

This Appendix accompanies Sections 3 of the main paper. It explores the space of optimization algorithms and hyperparameters for the Conv-2, Conv-4, and Conv-6 architectures evaluated in Section 3 with the same two purposes as Appendix G: explaining the hyperparameters used in the main body of the paper and evaluating the lottery ticket experiment on other choices of hyperparameters.

### H.1  EXPERIMENTAL METHODOLOGY

The Conv-2, Conv-4, and Conv-6 architectures are variants of the VGG (Simonyan & Zisserman, 2014) network architecture scaled down for the CIFAR10 (Krizhevsky & Hinton, 2009) dataset. Like VGG, the networks consist of a series of modules. Each module has two layers of 3x3 convolutional filters followed by a maxpool layer with stride 2. After all of the modules are two fully-connected layers of size 256 followed by an output layer of size 10; in VGG, the fully-connected layers are of size 4096 and the output layer is of size 1000. Like VGG, the first module has 64 convolutions in each layer, the second has 128, the third has 256, etc. The Conv-2, Conv-4, and Conv-6 architectures have 1, 2, and 3 modules, respectively.

The CIFAR10 dataset consists of 50,000 32x32 color (three-channel) training examples and 10,000 test examples. We randomly sampled a 5,000-example validation set from the training set and used the remaining 45,000 training examples as our training set for the rest of the paper. The hyperparameter selection experiments throughout this Appendix are evaluated on the validation set, and the examples in the main body of this paper (which make use of these hyperparameters) are evaluated on test set. The training set is presented to the network in mini-batches of 60 examples; at each epoch, the entire training set is shuffled.

The Conv-2, Conv-4, and Conv-6 networks are initialized with Gaussian Glorot initialization (Glorot & Bengio, 2010) and are trained for the number of iterations specified in Figure 2. The number of training iterations was selected such that heavily-pruned networks could still train in the time provided. On dropout experiments, the number of training iterations is tripled to provide enough time for the dropout-regularized networks to train. We optimize these networks with Adam, and select the learning rate for each network in this Appendix.

As with the MNIST experiments, validation and test performance is only considered retroactively and has no effect on the progression of the lottery ticket experiments. We measure validation and test loss and accuracy every 100 training iterations.

Each line in each graph of this section represents the average of three separate experiments, with error bars indicating the minimum and maximum value that any experiment took on at that point. (Experiments in the main body of the paper are conducted five times.)

We allow convolutional layers and fully-connected layers to be pruned at different rates; we select those rates for each network in this Appendix. The output layer is pruned at half of the rate of the fully-connected layers for the reasons described in Appendix G.

### H.2  LEARNING RATE

In this Subsection, we perform the lottery ticket experiment on the the Conv-2, Conv-4, and Conv-6 architectures as optimized with Adam at various learning rates.

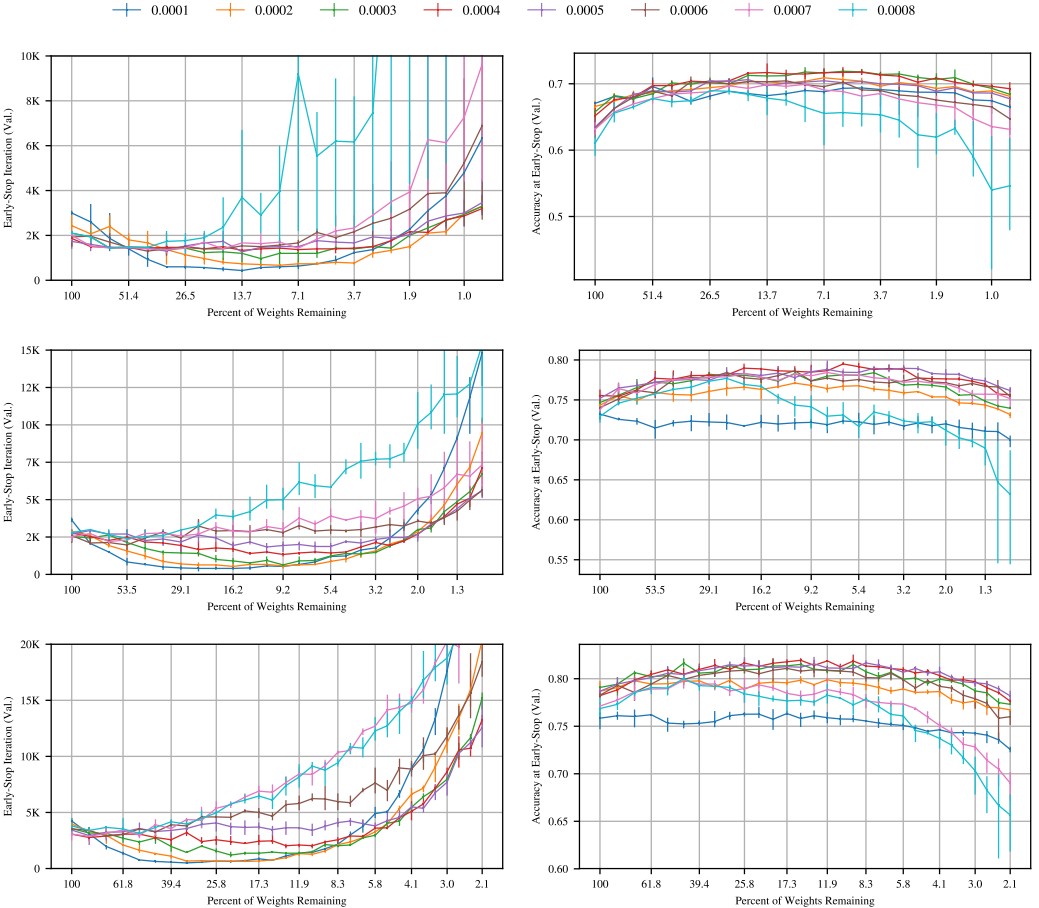

Figure 32: The early-stopping iteration and validation accuracy at that iteration of the iterative lottery ticket experiment on the Conv-2 (top), Conv-4 (middle), and Conv-6 (bottom) architectures trained using the Adam optimizer at various learning rates. Each line represents a different learning rate.

Here, we select the learning rate that we use for Adam in the main body of the paper. Our criteria for selecting the learning rate are the same as in Appendix G: minimizing training iterations and maximizing accuracy at early-stopping, finding winning tickets containing as few parameters as possible, and remaining conservative enough to apply to a range of other experiments.

Figure 32 shows the results of performing the iterative lottery ticket experiment on the Conv-2 (top), Conv-4 (middle), and Conv-6 (bottom) architectures. Since we have not yet selected the pruning rates for each network, we temporarily pruned fully-connected layers at 20% per iteration, convolutional layers at 10% per iteration, and the output layer at 10% per iteration; we explore this part of the hyperparameter space in a later subsection.

For Conv-2, we select a learning rate of 0.0002, which has the highest initial validation accuracy, maintains both high validation accuracy and low early-stopping times for the among the longest, and reaches the fastest early-stopping times. This learning rate also leads to a 3.3 percentage point improvement in validation accuracy when the network is pruned to 3% of its original size. Other learning rates, such 0.0004, have lower initial validation accuracy (65.2% vs 67.6%) but eventually reach higher absolute levels of validation accuracy (71.7%, a 6.5 percentage point increase, vs. 70.9%, a 3.3 percentage point increase). However, learning rate 0.0002 shows the highest proportional decrease in early-stopping times: 4.8x (when pruned to 8.8% of the original network size).

For Conv-4, we select learning rate 0.0003, which has among the highest initial validation accuracy, maintains high validation accuracy and fast early-stopping times when pruned by among the most, and balances improvements in validation accuracy (3.7 percentage point improvement to 78.6% when 5.4% of weights remain) and improvements in early-stopping time (4.27x when 11.1% of weights remain). Other learning rates reach higher validation accuracy (0.0004—3.6 percentage point improvement to 79.1% accuracy when 5.4% of weights remain) or show better improvements in early-stopping times (0.0002—5.1x faster when 9.2% of weights remain) but not both.

For Conv-6, we also select learning rate 0.0003 for similar reasons to those provided for Conv-4. Validation accuracy improves by 2.4 percentage points to 81.5% when 9.31% of weights remain and early-stopping times improve by 2.61x when pruned to 11.9%. Learning rate 0.0004 reaches high final validation accuracy (81.9%, an increase of 2.7 percentage points, when 15.2% of weights remain) but with smaller improvements in early-stopping times, and learning rate 0.0002 shows greater improvements in early-stopping times (6.26x when 19.7% of weights remain) but reaches lower overall validation accuracy.

We note that, across nearly all combinations of learning rates, the lottery ticket pattern—where early-stopping times were maintain or decreased and validation accuracy was maintained or increased during the course of the lottery ticket experiment—continues to hold. This pattern fails to hold at the very highest learning rates: early-stopping times decreased only briefly (in the case of Conv-2 or Conv-4) or not at all (in the case of Conv-6), and accuracy increased only briefly (in the case of all three networks). This pattern is similar to that which we observe in Section 4: at the highest learning rates, our iterative pruning algorithm fails to find winning tickets.

## H.3 OTHER OPTIMIZATION ALGORITHMS

### H.3.1 SGD

Here, we explore the behavior of the lottery ticket experiment when the Conv-2, Conv-4, and Conv-6 networks are optimized with stochastic gradient descent (SGD) at various learning rates. The results of doing so appear in Figure 33. In general, these networks—particularly Conv-2 and Conv-4—proved challenging to train with SGD and Glorot initialization. As Figure 33 reflects, we could not find SGD learning rates for which the unpruned networks matched the validation accuracy of the same networks when trained with Adam; at best, the SGD-trained unpruned networks were typically 2-3 percentage points less accurate. At higher learning rates than those in Figure 32, gradients tended to explode when training the unpruned network; at lower learning rates, the networks often failed to learn at all.

At all of the learning rates depicted, we found winning tickets. In all cases, early-stopping times initially decreased with pruning before eventually increasing again, just as in other lottery ticket experiments. The Conv-6 network also exhibited the same accuracy patterns as other experiments, with validation accuracy initially increasing with pruning before eventually decreasing again.

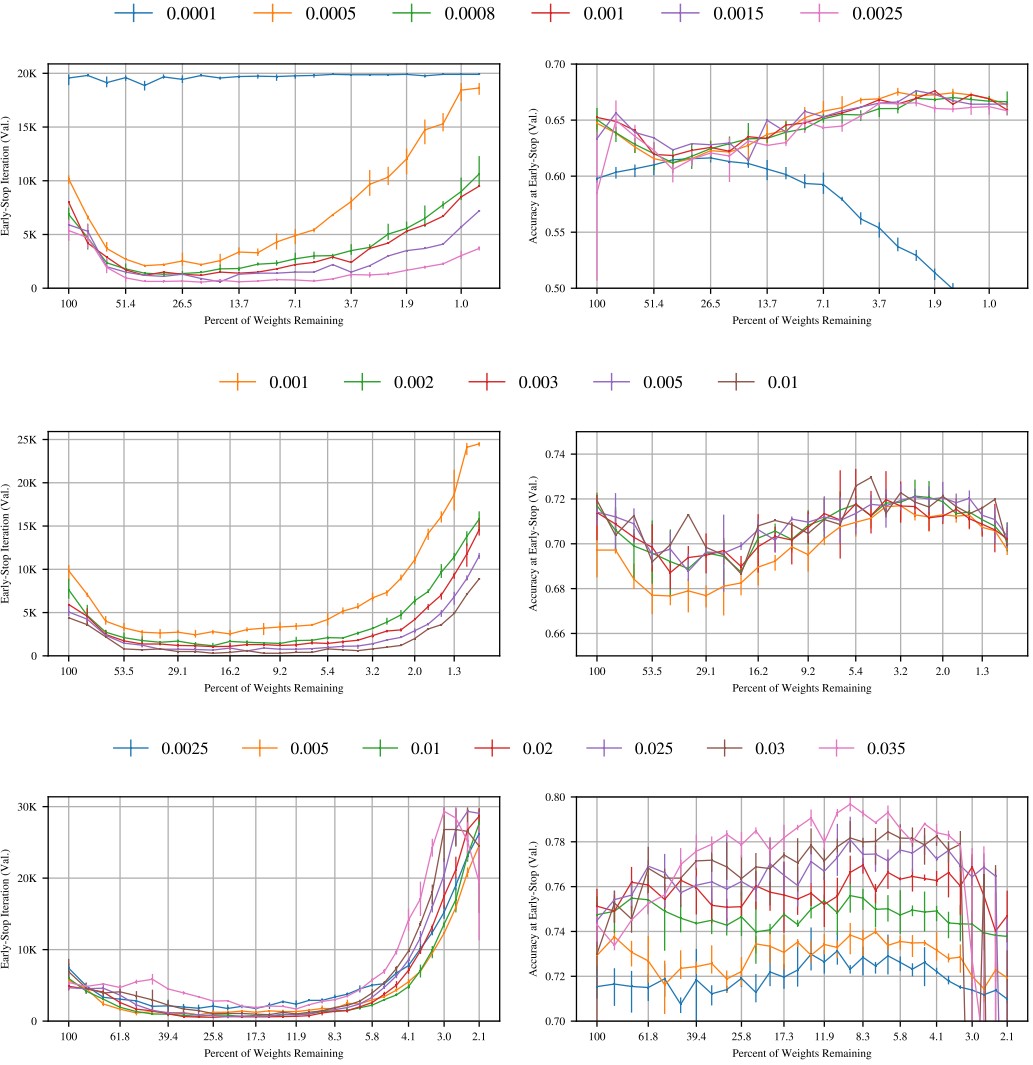

Figure 33: The early-stopping iteration and validation accuracy at that iteration of the iterative lottery ticket experiment on the Conv-2 (top), Conv-4 (middle), and Conv-6 (bottom) architectures trained using SGD at various learning rates. Each line represents a different learning rate. The legend for each pair of graphs is above the graphs.

However, the Conv-2 and Conv-4 architectures exhibited a different validation accuracy pattern from other experiments in this paper. Accuracy initially declined with pruning before rising as the network was further pruned; it eventually matched or surpassed the accuracy of the unpruned network. When they eventually did surpass the accuracy of the original network, the pruned networks reached early-stopping in about the same or fewer iterations than the original network, constituting a winning ticket by our definition. Interestingly, this pattern also appeared for Conv-6 networks at slower SGD learning rates, suggesting that faster learning rates for Conv-2 and Conv-4 than those in Figure 32 might cause the usual lottery ticket accuracy pattern to reemerge. Unfortunately, at these higher learning rates, gradients exploded on the unpruned networks, preventing us from running these experiments.

### H.3.2 MOMENTUM

Here, we explore the behavior of the lottery ticket experiment when the network is optimized with SGD with momentum (0.9) at various learning rates. The results of doing so appear in Figure 34. In general, the lottery ticket pattern continues to apply, with early-stopping times decreasing and accuracy increasing as the networks are pruned. However, there were two exceptions to this pattern:

1. At the very lowest learning rates (e.g., learning rate 0.001 for Conv-4 and all but the highest learning rate for Conv-2), accuracy initially decreased before increasing to higher levels than reached by the unpruned network; this is the same pattern we observed when training these networks with SGD.

2. At the very highest learning rates (e.g., learning rates 0.005 and 0.008 for Conv-2 and Conv-4), early-stopping times never decreased and instead remained stable before increasing; this is the same pattern we observed for the highest learning rates when training with Adam.

### H.4 ITERATIVE PRUNING RATE

For the convolutional network architectures, we select different pruning rates for convolutional and fully-connected layers. In the Conv-2 and Conv-4 architectures, convolutional parameters make up a relatively small portion of the overall number of parameters in the models. By pruning convolutions more slowly, we are likely to be able to prune the model further while maintaining performance. In other words, we hypothesize that, if all layers were pruned evenly, convolutional layers would become a bottleneck that would make it more difficult to find lower parameter-count models that are still able to learn. For Conv-6, the opposite may be true: since nearly two thirds of its parameters are in convolutional layers, pruning fully-connected layers could become the bottleneck.

Our criterion for selecting hyperparameters in this section is to find a combination of pruning rates that allows networks to reach the lowest possible parameter-counts while maintaining validation accuracy at or above the original accuracy and early-stopping times at or below that for the original network.

Figure 35 shows the results of performing the iterative lottery ticket experiment on Conv-2 (top), Conv-4 (middle), and Conv-6 (bottom) with different combinations of pruning rates.

According to our criteria, we select an iterative convolutional pruning rate of 10% for Conv-2, 10% for Conv-4, and 15% for Conv-6. For each network, any rate between 10% and 20% seemed reasonable. Across all convolutional pruning rates, the lottery ticket pattern continued to appear.

### H.5 LEARNING RATES (DROPOUT)

In order to train the Conv-2, Conv-4, and Conv-6 architectures with dropout, we repeated the exercise from Section H.2 to select appropriate learning rates. Figure 32 shows the results of performing the iterative lottery ticket experiment on Conv-2 (top), Conv-4 (middle), and Conv-6 (bottom) with dropout and Adam at various learning rates. A network trained with dropout takes longer to learn, so we trained each architecture for three times as many iterations as in the experiments without dropout: 60,000 iterations for Conv-2, 75,000 iterations for Conv-4, and 90,000 iterations for Conv-6. We iteratively pruned these networks at the rates determined in Section H.4.

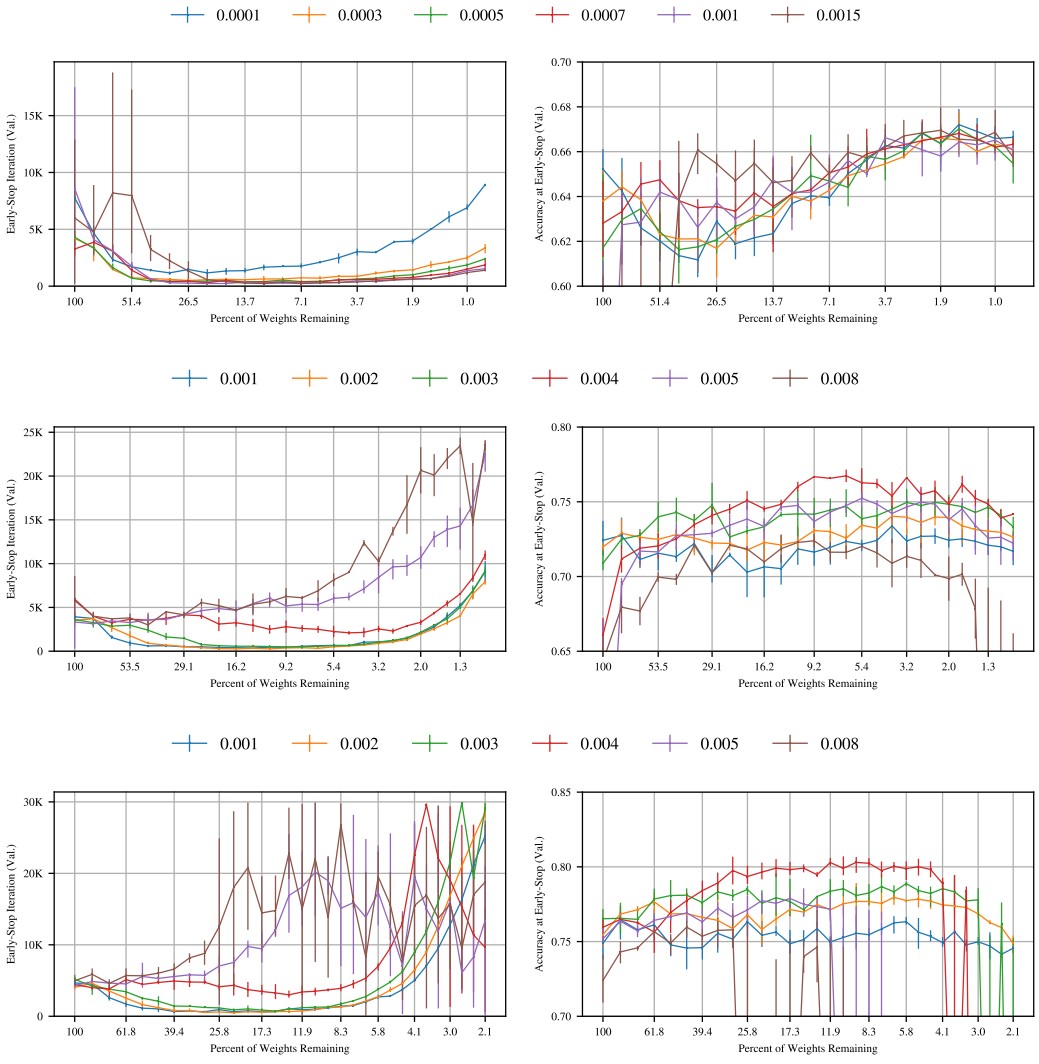

Figure 34: The early-stopping iteration and validation accuracy at that iteration of the iterative lottery ticket experiment on the Conv-2 (top), Conv-4 (middle), and Conv-6 (bottom) architectures trained using SGD with momentum (0.9) at various learning rates. Each line represents a different learning rate. The legend for each pair of graphs is above the graphs. Lines that are unstable and contain large error bars (large vertical lines) indicate that some experiments failed to learn effectively, leading to very low accuracy and very high early-stopping times; these experiments reduce the averages that the lines trace and lead to much wider error bars.

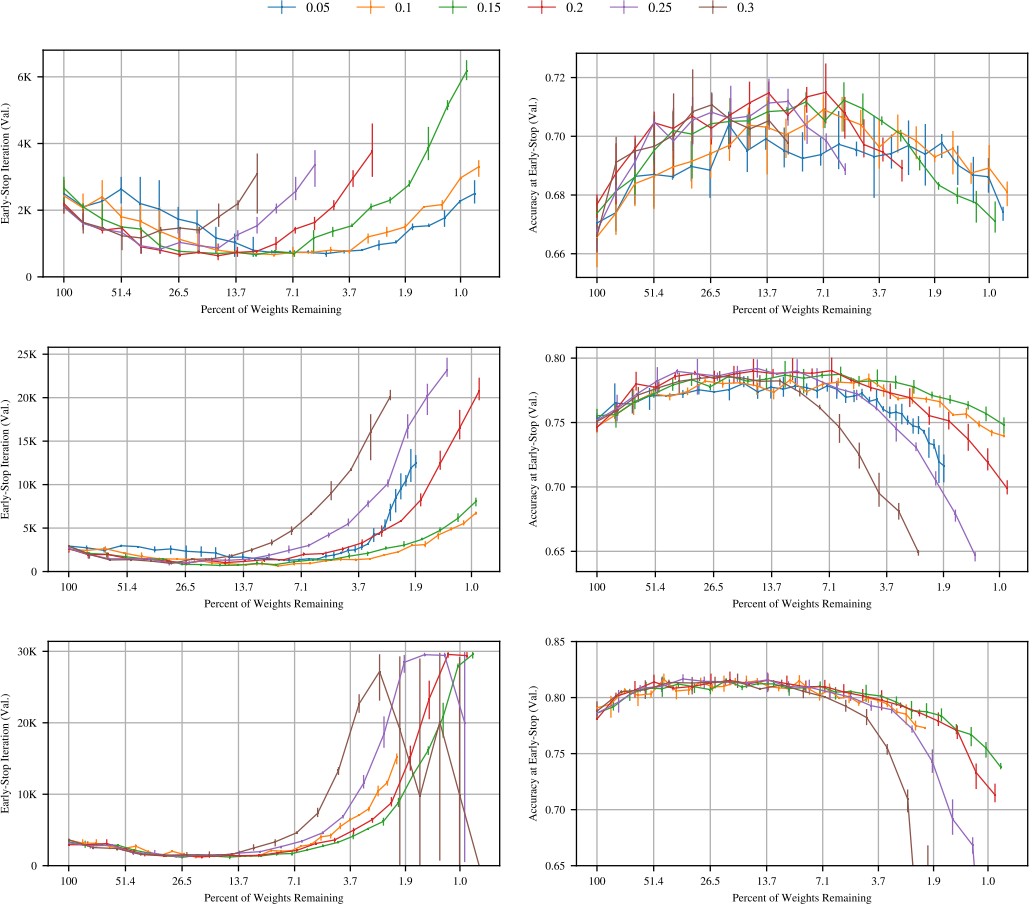

Figure 35: The early-stopping iteration and validation accuracy at that iteration of the iterative lottery ticket experiment on the Conv-2 (top), Conv-4 (middle), and Conv-6 (bottom) architectures with an iterative pruning rate of 20% for fully-connected layers. Each line represents a different iterative pruning rate for convolutional layers.

The Conv-2 network proved to be difficult to consistently train with dropout. The top right graph in Figure 36 contains wide error bars and low average accuracy for many learning rates, especially early in the lottery ticket experiments. This indicates that some or all of the training runs failed to learn; when they were averaged into the other results, they produced the aforementioned pattern in the graphs. At learning rate 0.0001, none of the three trials learned productively until pruned to more than 26.5%, at which point all three trials started learning. At learning rate 0.0002, some of the trials failed to learn productively until several rounds of iterative pruning had passed. At learning rate 0.0003, all three networks learned productively at every pruning level. At learning rate 0.0004, one network occasionally failed to learn. We selected learning rate 0.0003, which seemed to allow networks to learn productively most often while achieving among the highest initial accuracy.

It is interesting to note that networks that were unable to learn at a particular learning rate (for example, 0.0001) eventually began learning after several rounds of the lottery ticket experiment (that is, training, pruning, and resetting repeatedly). It is worth investigating whether this phenomenon was entirely due to pruning (that is, removing any random collection of weights would put the network in a configuration more amenable to learning) or whether training the network provided useful information for pruning, even if the network did not show improved accuracy.

For both the Conv-4 and Conv-6 architectures, a slightly slower learning rate (0.0002 as opposed to 0.0003) leads to the highest accuracy on the unpruned networks in addition to the highest sustained accuracy and fastest sustained learning as the networks are pruned during the lottery ticket experiment.

With dropout, the unpruned Conv-4 architecture reaches an average validation accuracy of 77.6%, a 2.7 percentage point improvement over the unpruned Conv-4 network trained without dropout and one percentage point lower than the highest average validation accuracy attained by a winning ticket. The dropout-trained winning tickets reach 82.6% average validation accuracy when pruned to 7.6%. Early-stopping times improve by up to 1.58x (when pruned to 7.6%), a smaller improvement than then 4.27x achieved by a winning ticket obtained without dropout.

With dropout, the unpruned Conv-6 architecture reaches an average validation accuracy of 81.3%, an improvement of 2.2 percentage points over the accuracy without dropout; this nearly matches the 81.5% average accuracy obtained by Conv-6 trained without dropout and pruned to 9.31%. The dropout-trained winning tickets further improve upon these numbers, reaching 84.8% average validation accuracy when pruned to 10.5%. Improvements in early-stopping times are less dramatic than without dropout: a 1.5x average improvement when the network is pruned to 15.1%.

At all learning rates we tested, the lottery ticket pattern generally holds for accuracy, with improvements as the networks are pruned. However, not all learning rates show the decreases in early-stopping times. To the contrary, none of the learning rates for Conv-2 show clear improvements in early-stopping times as seen in the other lottery ticket experiments. Likewise, the faster learning rates for Conv-4 and Conv-6 maintain the original early-stopping times until pruned to about 40%, at which point early-stopping times steadily increase.

### H.6 PRUNING CONVOLUTIONS VS. PRUNING FULLY-CONNECTED LAYERS

Figure 37 shows the effect of pruning convolutions alone (green), fully-connected layers alone (orange) and pruning both (blue). The x-axis measures the number of parameters remaining to emphasize the relative contributions made by pruning convolutions and fully-connected layers to the overall network. In all three cases, pruning convolutions alone leads to higher test accuracy and faster learning; pruning fully-connected layers alone generally causes test accuracy to worsen and learning to slow. However, pruning convolutions alone has limited ability to reduce the overall parameter-count of the network, since fully-connected layers comprise 99%, 89%, and 35% of the parameters in Conv-2, Conv-4, and Conv-6.

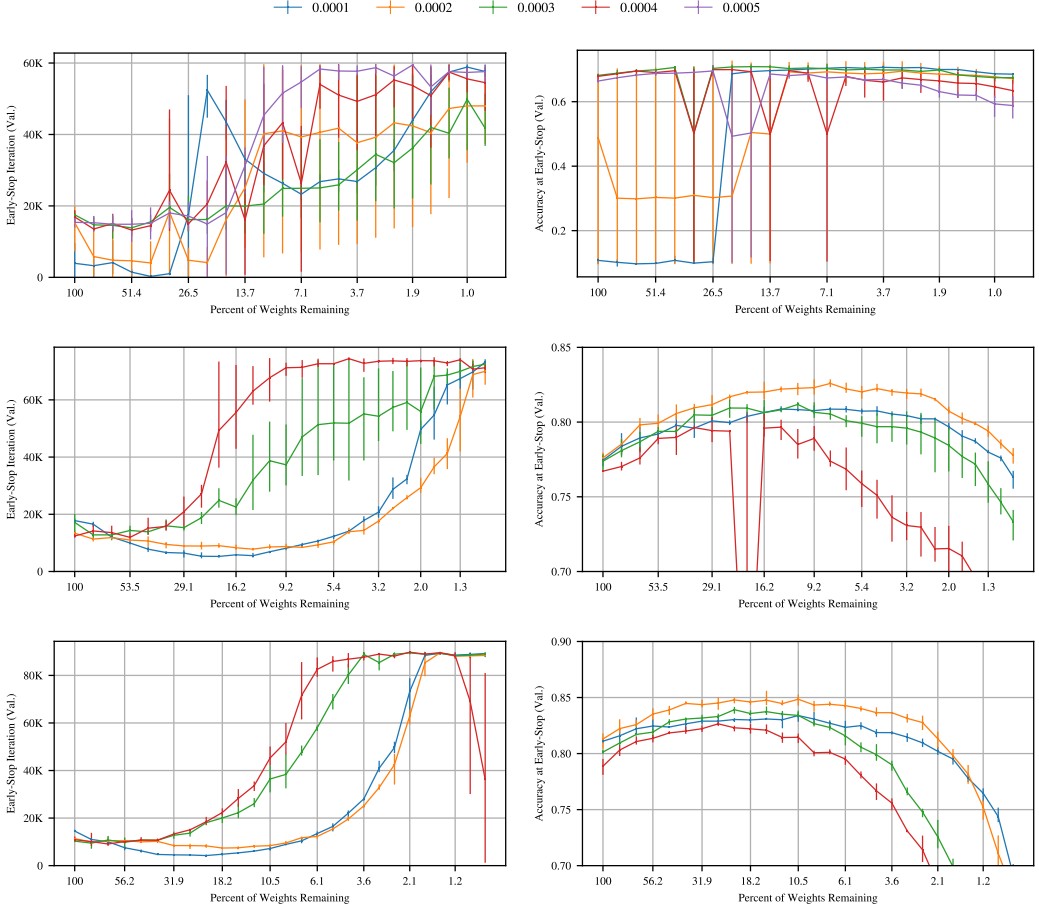

Figure 36: The early-stopping iteration and validation accuracy at that iteration of the iterative lottery ticket experiment on the Conv-2 (top), Conv-4 (middle), and Conv-6 (bottom) architectures trained using dropout and the Adam optimizer at various learning rates. Each line represents a different learning rate.

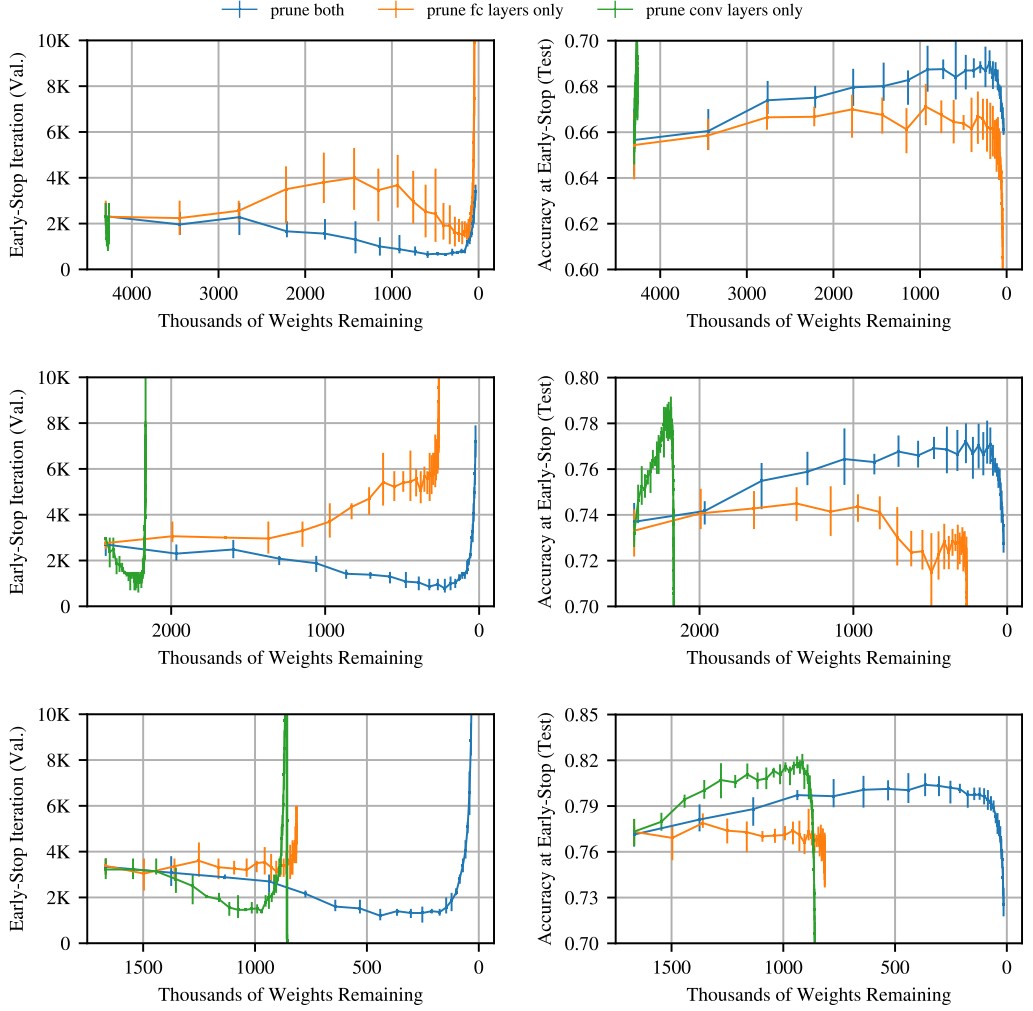

Figure 37: Early-stopping iteration and accuracy of the Conv-2 (top), Conv-4 (middle), and Conv-6 (bottom) networks when only convolutions are pruned, only fully-connected layers are pruned, and both are pruned. The x-axis measures the number of parameters remaining, making it possible to see the relative contributions to the overall network made by pruning FC layers and convolutions individually.

# I HYPERPARAMETER EXPLORATION FOR VGG-19 AND RESNET-18 ON CIFAR10

This Appendix accompanies the VGG-19 and Resnet-18 experiments in Section 4. It details the pruning scheme, training regimes, and hyperparameters that we use for these networks.

## I.1 GLOBAL PRUNING

In our experiments with the Lenet and Conv-2/4/6 architectures, we separately prune a fraction of the parameters in each layer (*layer-wise pruning*). In our experiments with VGG-19 and Resnet-18, we instead prune *globally*; that is, we prune all of the weights in convolutional layers collectively without regard for the specific layer from which any weight originated.

Figures 38 (VGG-19) and 39 (Resnet-18) compare the winning tickets found by global pruning (solid lines) and layer-wise pruning (dashed lines) for the hyperparameters from Section 4. When training VGG-19 with learning rate 0.1 and warmup to iteration 10,000, we find winning tickets when $P_m \geq 6.9\%$ for layer-wise pruning vs. $P_m \geq 1.5\%$ for global pruning. For other hyperparameters, accuracy similarly drops off when sooner for layer-wise pruning than for global pruning. Global pruning also finds smaller winning tickets than layer-wise pruning for Resnet-18, but the difference is less extreme than for VGG-19.

In Section 4, we discuss the rationale for the efficacy of global pruning on deeper networks. In summary, the layers in these deep networks have vastly different numbers of parameters (particularly severely so for VGG-19); if we prune layer-wise, we conjecture that layers with fewer parameters become bottlenecks on our ability to find smaller winning tickets.

Regardless of whether we use layer-wise or global pruning, the patterns from Section 4 hold: at learning rate 0.1, iterative pruning finds winning tickets for neither network; at learning rate 0.01, the lottery ticket pattern reemerges; and when training with warmup to a higher learning rate, iterative pruning finds winning tickets. Figures 40 (VGG-19) and 41 (Resnet-18) present the same data as Figures 7 (VGG-19) and 8 (Resnet-18) from Section 4 with layer-wise pruning rather than global pruning. The graphs follow the same trends as in Section 4, but the smallest winning tickets are larger than those found by global pruning.

## I.2 VGG-19 DETAILS

The VGG19 architecture was first designed by Simonyan & Zisserman (2014) for Imagenet. The version that we use here was adapted by Liu et al. (2019) for CIFAR10. The network is structured as described in Figure 2: it has five groups of 3x3 convolutional layers, the first four of which are followed by max-pooling (stride 2) and the last of which is followed by average pooling. The network has one final dense layer connecting the result of the average-pooling to the output.

We largely follow the training procedure for resnet18 described in Appendix I:

- We use the same train/test/validation split.
- We use the same data augmentation procedure.
- We use a batch size of 64.
- We use batch normalization.
- We use a weight decay of 0.0001.
- We use three stages of training at decreasing learning rates. We train for 160 epochs (112,480 iterations), decreasing the learning rate by a factor of ten after 80 and 120 epochs.
- We use Gaussian Glorot initialization.

We globally prune the convolutional layers of the network at a rate of 20% per iteration, and we do not prune the 5120 parameters in the output layer.

Liu et al. (2019) uses an initial pruning rate of 0.1. We train VGG19 with both this learning rate and a learning rate of 0.01.

## I.3 RESNET-18 DETAILS

The Resnet-18 architecture was first introduced by He et al. (2016). The architecture comprises 20 total layers as described in Figure 2: a convolutional layer followed by nine pairs of convolutional layers (with residual connections around the pairs), average pooling, and a fully-connected output layer.

We follow the experimental design of He et al. (2016):

- We divide the training set into 45,000 training examples and 5,000 validation examples. We use the validation set to select hyperparameters in this appendix and the test set to evaluate in Section 4.
- We augment training data using random flips and random four pixel pads and crops.
- We use a batch size of 128.
- We use batch normalization.
- We use weight decay of 0.0001.
- We train using SGD with momentum (0.9).
- We use three stages of training at decreasing learning rates. Our stages last for 20,000, 5,000, and 5,000 iterations each, shorter than the 32,000, 16,000, and 16,000 used in He et al. (2016). Since each of our iterative pruning experiments requires training the network 15-30 times consecutively, we select this abbreviated training schedule to make it possible to explore a wider range of hyperparameters.
- We use Gaussian Glorot initialization.

We globally prune convolutions at a rate of 20% per iteration. We do not prune the 2560 parameters used to downsample residual connections or the 640 parameters in the fully-connected output layer, as they comprise such a small portion of the overall network.

## I.4 LEARNING RATE

In Section 4, we observe that iterative pruning is unable to find winning tickets for VGG-19 and Resnet-18 at the typical, high learning rate used to train the network (0.1) but it is able to do so at a lower learning rate (0.01). Figures 42 and 43 explore several other learning rates. In general, iterative pruning cannot find winning tickets at any rate above 0.01 for either network; for higher learning rates, the pruned networks with the original initialization perform no better than when randomly reinitialized.

## I.5 WARMUP ITERATION

In Section 4, we describe how adding linear warmup to the initial learning rate makes it possible to find winning tickets for VGG-19 and Resnet-18 at higher learning rates (and, thereby, winning tickets that reach higher accuracy). In Figures 44 and 45, we explore the number of iterations $k$ over which warmup should occur.

For VGG-19, we were able to find values of $k$ for which iterative pruning could identify winning tickets when the network was trained at the original learning rate (0.1). For Resnet-18, warmup made it possible to increase the learning rate from 0.01 to 0.03, but no further. When exploring values of $k$, we therefore us learning rate 0.1 for VGG-19 and 0.03 for Resnet-18.

In general, the greater the value of $k$, the higher the accuracy of the eventual winning tickets.

**Resnet-18.** For values of $k$ below 5000, accuracy improves rapidly as $k$ increases. This relationship reaches a point of diminishing returns above $k = 5000$. For the experiments in Section 4, we select $k = 20000$, which achieves the highest validation accuracy.

**VGG-19.** For values of $k$ below 5000, accuracy improves rapidly as $k$ increases. This relationship reaches a point of diminishing returns above $k = 5000$. For the experiments in Section 4, we select $k = 10000$, as there is little benefit to larger values of $k$.

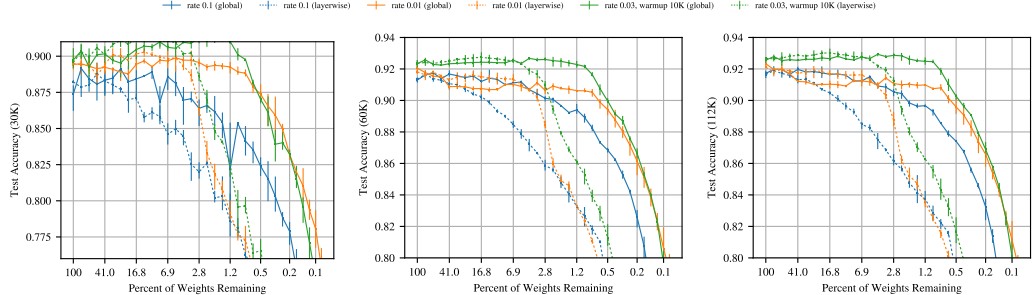

Figure 38: Validation accuracy (at 30K, 60K, and 112K iterations) of VGG-19 when iteratively pruned with global (solid) and layer-wise (dashed) pruning.

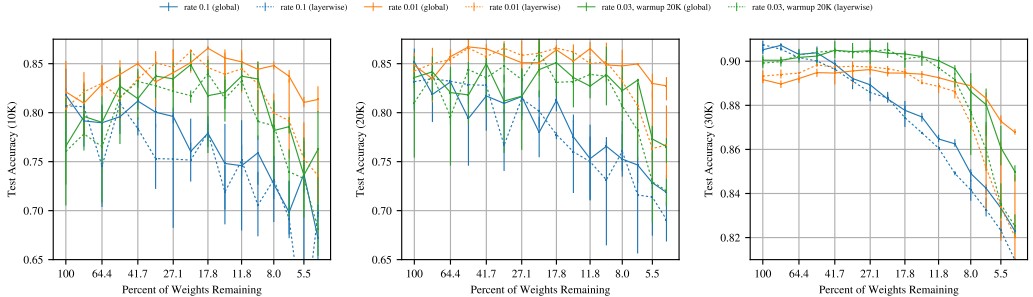

Figure 39: Validation accuracy (at 10K, 20K, and 30K iterations) of Resnet-18 when iteratively pruned with global (solid) and layer-wise (dashed) pruning.

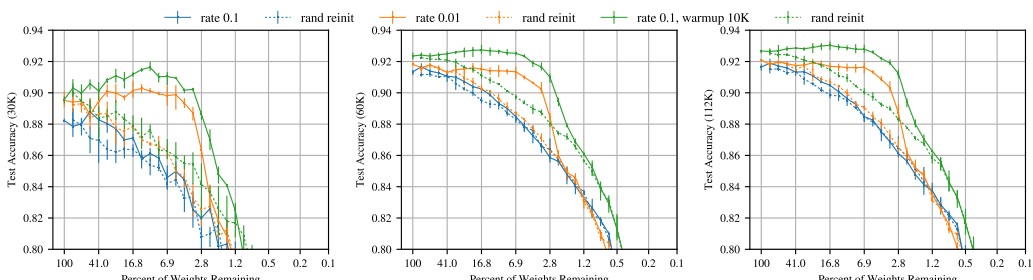

Figure 40: Test accuracy (at 30K, 60K, and 112K iterations) of VGG-19 when iteratively pruned with layer-wise pruning. This is the same as Figure 7, except with layer-wise pruning rather than global pruning.

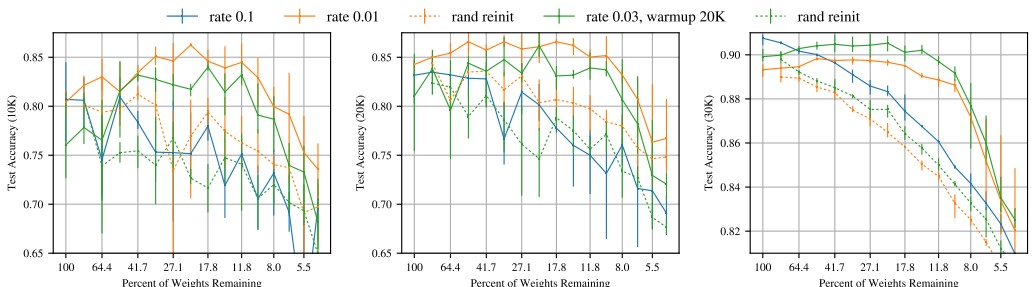

Figure 41: Test accuracy (at 10K, 20K, and 30K iterations) of Resnet-18 when iteratively pruned with layer-wise pruning. This is the same as Figure 8 except with layer-wise pruning rather than global pruning.

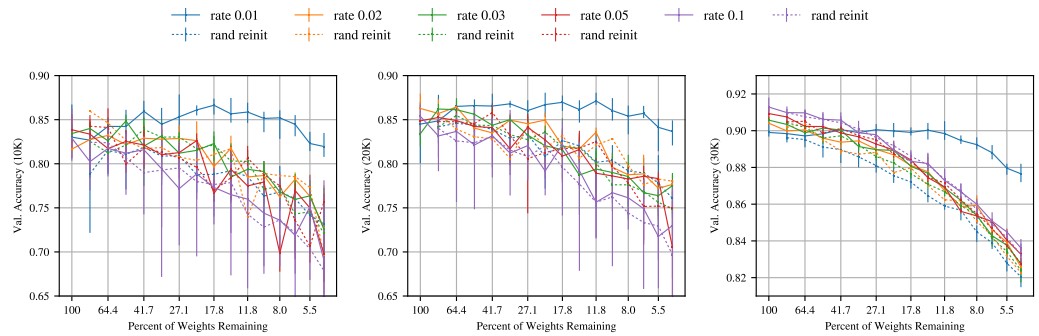

Figure 42: Validation accuracy (at 10K, 20K, and 30K iterations) of Resnet-18 when iteratively pruned and trained with various learning rates.

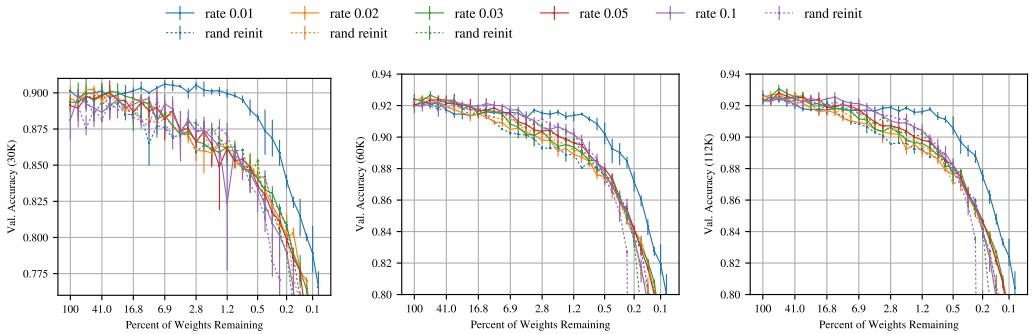

Figure 43: Validation accuracy (at 30K, 60K, and 112K iterations) of VGG-19 when iteratively pruned and trained with various learning rates.

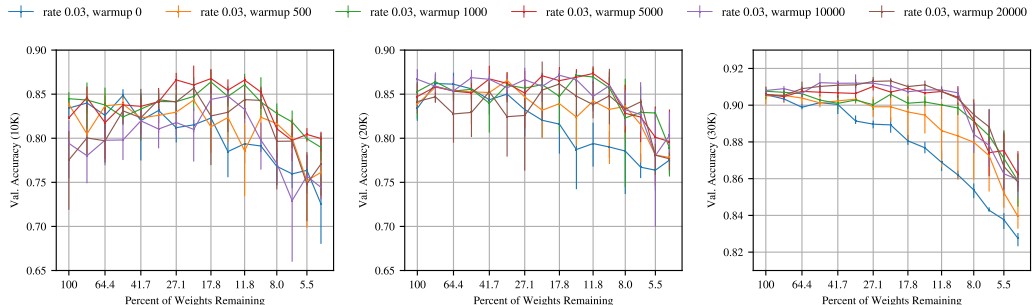

Figure 44: Validation accuracy (at 10K, 20K, and 30K iterations) of Resnet-18 when iteratively pruned and trained with varying amounts of warmup at learning rate 0.03.

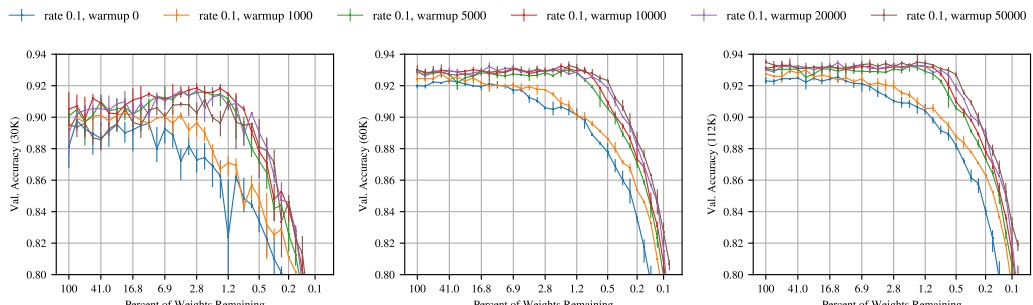

Figure 45: Validation accuracy (at 30K, 60K, and 112K iterations) of VGG-19 when iteratively pruned and trained with varying amounts of warmup at learning rate 0.1.

