# OpenReview forum: "The Lottery Ticket Hypothesis: Finding Sparse, Trainable Neural Networks"
_ICLR.cc/2019/Conference_

### Official Review · AnonReviewer3 · 2018-10-31
**Intriguing results that challenge the common understanding of how neural network training works**

**Rating:** 9
**Confidence:** 4

**Review:**

(Score raised from 8 to 9 after rebuttal)
The paper examines the hypothesis that randomly initialized (feed-forward) neural networks contain sub-networks that train well in the sense that they converge equally fast or faster and reach the same or better classification accuracy. Interestingly, such sub-networks can be identified by simple, magnitude-based pruning. It is crucial that these sub-networks are initialized with their original initialization values, otherwise they typically fail to be trained, implying that it is not purely the structure of the sub-networks that matters. The paper thoroughly investigates the existence of such “winning-tickets” on MNIST and CIFAR-10 on both, fully connected but also convolutional neural networks. Winning-tickets are found across networks, various optimizers, at different pruning-levels and across various other hyper-parameters. The experiments also show that iterative pruning (with re-starts) is more effective at finding winning-tickets.

The paper adds a novel and interesting angle to the question of why neural networks apparently need to be heavily over-parameterized for training. This question is intriguing and of high importance to further the understanding of how neural networks train. Additionally, the findings might have practical relevance as they might help avoid unnecessary over-parameterization which, in turn, might save use of computational resources and energy. The main idea is simple (which is good) and can be tested with relatively simple experiments (also good). The experiments conducted in the paper are clean (averaging over multiple runs, controlling for a lot of factors) and should allow for easy reproduction but also for clean comparison against future experiments. The experimental section is well executed, the writing is clear and good and related work is taken into account to a sufficient degree. The paper touches upon a very intriguing “feature” of neural networks and, in my opinion, should be relevant to theorists and practitioners across many sub-fields of deep learning research. I therefore vote and argue for accepting the paper for presentation at the conference. The following comments are suggestions to the authors on how to further improve the paper. I do not expect all issues to be addressed in the camera-ready version.

1) The main “weakness” of the paper might be that, while the amount of experiments and controls is impressive, the generality of the lottery ticket hypothesis remains somewhat open. Even when restricting the statement to feed-forward networks only, the networks investigated in the paper are relatively “small” and MNIST and CIFAR-10 bear the risk of finding patterns that do not hold when scaling to larger-scale networks and tasks. I acknowledge and support the author’s decision to have thorough and clean experiments on these small models and tasks, rather than having half-baked results on ImageNet, etc. The downside of this is that the experiments are thus not sufficient to claim (with reasonable certainty) that the lottery ticket hypothesis holds “in general”. The paper would be stronger, if the existence of winning tickets on larger-scale experiments or tasks other than classification were shown - even if these experiments did not have a large number of control experiments/ablation studies.

2)  While the paper shows the existence of winning tickets robustly and convincingly on the networks/tasks investigated, the next important question would be how to systematically and reliably “break” the existence of lottery tickets. Can they be attributed to a few fundamental factors? Are they a consequence of batch-wise, gradient-based optimization, or an inherent feature of neural networks, or is it the loss functions commonly used, …? On page 2, second paragraph, the paper states: ”When randomly reinitialized, our winning tickets no longer match the performance of the original network, explaining the difficulty of training pruned networks from scratch”. I don’t fully agree - the paper certainly sheds some light on the issue, but an actual explanation would result in a testable hypothesis. My comment here is intended to be constructive criticism, I think that the paper has enough “juice” and novelty for being accepted - I am merely pointing out that the overall story is not yet conclusive (and I am aware that it might need several more publications to find these answers).

3) Do the winning tickets generalize across hyper-parameters or even tasks. I.e. if a winning ticket is found with one set of hyper-parameters, but then Optimizer/learning-rate/etc. are changed, does the winning-ticket still lead to improved convergence and accuracy? Same question for data-sets: do winning-tickets found on CIFAR-100 also work for CIFAR-10 and vice versa? If winning-tickets turn out to generalize well, in the extreme this could allow “shipping” each network architecture with a few good winning-tickets, thus making it unnecessary to apply expensive iterative pruning every time. I would not expect generalization across data-sets, but it would be highly interesting to see if winning tickets generalize in any way (after all I am still surprised by how well adversarial examples generalize and transfer).

4) Some things that would be interesting to try:
4a) Is there anything special about the pruned/non-pruned weights at the time of initialization? Did they start out with very small values already or are they all “behind” some (dead) downstream neuron? Is there anything that might essentially block gradient signal from updating the pruned neurons? This could perhaps be checked by recording weights’ “trajectories” during training to see if there is a correlation between the “distance weights traveled” and whether or not they end up in the winning ticket.
4b) Do ARD-style/Bayesian approaches or second-order methods to pruning identify (roughly) the same neurons for pruning?

5) Typo (should be through): “we find winning tickets though a principled search process”

6) For the standard ConvNets I assume you did not use batchnorm. Does batchnorm interfere in any way with the existence of winning tickets? (at least on ResNet they seem to exist with batchnorm as well)

---

> ### Author Response · Authors · 2018-11-26
> **Author Response (Part 2)**
>
>
> > 4. Some things that would be interesting to try: 4a) Is there anything special about the pruned/non-pruned weights at the time of initialization? Did they start out with very small values already or are they all “behind” some (dead) downstream neuron? Is there anything that might essentially block gradient signal from updating the pruned neurons? This could perhaps be checked by recording weights’ “trajectories” during training to see if there is a correlation between the “distance weights traveled” and whether or not they end up in the winning ticket.
>
> In the new Appendix D, we study the pruned and non-pruned weights at the time of initialization. We find that winning ticket initializations tend to come from the extremes of the truncated normal distribution from which the unpruned networks are initialized. We are interested in studying the other questions you mention in future work. We also look at the distance weights travel in the unpruned network, finding that weights that are part of the eventual winning tickets tend to move more than weights that are not part of the winning ticket.
>
> ---
>
> > 4b) Do ARD-style/Bayesian approaches or second-order methods to pruning identify (roughly) the same neurons for pruning?
>
> These are great questions that we are interested in understanding as well. In order to keep our experiments as simple and tractable as possible, we opted to focus on a single, simple, widely-accepted pruning method. However, we have updated our limitations section (Section 7) to reflect that we only use a single identification technique and that other techniques may produce winning tickets with different properties (e.g., fewer weights, improved training times, better generalization, or better performance on hardware).
>
> ---
>
> > 5. Typo (should be through): “we find winning tickets though a principled search process”
>
> Nice catch - it should now be corrected!
>
> ---
>
> > For the standard ConvNets I assume you did not use batchnorm. Does batchnorm interfere in any way with the existence of winning tickets? (at least on ResNet they seem to exist with batchnorm as well)
>
> The new networks (resnet18 and vgg16/19) all use batchnorm. You're correct that lenet and conv2/4/6 do not use batchnorm. As you note, since we still find winning tickets on these larger networks, it does not appear that batchnorm interferes with the existence of winning tickets.

---

> > ### Comment · AnonReviewer3 · 2018-11-29
> > **Response to Rebuttal**
> >
> > Thanks for the very detailed response, the additional experiments and analysis and the updated manuscript. I am particularly pleased to see the additional experiments (not that the original manuscript was lacking experimental results) and the analysis in Appendix D. I think that the current paper is "filled to the brink" with interesting experiments and results (which are conducted in a very solid fashion) - there are many interesting follow-up questions (quite a few of which have been named by the reviewers) and it is tempting to add even more results, but I agree with the authors that these questions deserve a separate publication.
> >
> > I also appreciate a more formal statement of the lottery-ticket hypothesis.
> >
> > The questions and issues raised in my review have all been addressed in a satisfactory fashion - the paper got even stronger. Looking forward to reading followup work on how well winning tickets generalize, whether they appear in non-classification tasks and whether other pruning methods identify the same winning tickets or not.

---

> ### Author Response · Authors · 2018-11-26
> **Author Response (Part 1)**
>
>
> Thank you so much for your thoughtful review. Below, you will find our responses to your questions and comments. We have modified the paper to reflect your feedback, and we are very interested in any further feedback you have about the new version of the paper.
>
> We have summarized the changes in the new version of the paper in a top-level comment called "Summary of Changes in the New Version."
>
> Where multiple reviewers made similar comments, we have grouped the answers into a "Common Questions" comment; you can find this comment as a response to our top-level comment called "Summary of Changes in the New Version."
>
> ---
>
> > I acknowledge and support the author’s decision to have thorough and clean experiments on these small models and tasks, rather than having half-baked results on ImageNet, etc. The downside of this is that the experiments are thus not sufficient to claim (with reasonable certainty) that the lottery ticket hypothesis holds “in general”. The paper would be stronger, if the existence of winning tickets on larger-scale experiments or tasks other than classification were shown - even if these experiments did not have a large number of control experiments/ablation studies.
>
> Please see Common Questions.
>
> ---
>
> > 2. While the paper shows the existence of winning tickets robustly and convincingly on the networks/tasks investigated, the next important question would be how to systematically and reliably “break” the existence of lottery tickets. Can they be attributed to a few fundamental factors?
>
> Please see Common Questions.
>
> ---
>
> > Are they a consequence of batch-wise, gradient-based optimization, or an inherent feature of neural networks, or is it the loss functions commonly used, …?
>
> In Appendices D and E, we show that the existence of winning tickets in lenet and conv2/4/6 is independent of the instantiation of a gradient-based optimization method (at least across Adam, SGD, and Momentum). However, we agree that there are still broader questions about the origin of winning tickets. We hope that the work in this paper makes it possible for us and others to follow with answers to these questions.
>
> ---
>
> > On page 2, second paragraph, the paper states: ”When randomly reinitialized, our winning tickets no longer match the performance of the original network, explaining the difficulty of training pruned networks from scratch”. I don’t fully agree - the paper certainly sheds some light on the issue, but an actual explanation would result in a testable hypothesis. My comment here is intended to be constructive criticism, I think that the paper has enough “juice” and novelty for being accepted - I am merely pointing out that the overall story is not yet conclusive (and I am aware that it might need several more publications to find these answers).
>
> This is an excellent observation and we have changed our language accordingly.
>
> ---
>
> > 3. Do the winning tickets generalize across hyper-parameters or even tasks. I.e. if a winning ticket is found with one set of hyper-parameters, but then Optimizer/learning-rate/etc. are changed, does the winning-ticket still lead to improved convergence and accuracy? Same question for data-sets: do winning-tickets found on CIFAR-100 also work for CIFAR-10 and vice versa? If winning-tickets turn out to generalize well, in the extreme this could allow “shipping” each network architecture with a few good winning-tickets, thus making it unnecessary to apply expensive iterative pruning every time. I would not expect generalization across data-sets, but it would be highly interesting to see if winning tickets generalize in any way (after all I am still surprised by how well adversarial examples generalize and transfer).
>
> This is a great question that we are interested in as well. We have conducted some exploratory experiments in each of these directions (changing hyperparameters and changing datasets) in preparation for future research, but the results are too preliminary to merit discussion. We have noted the dataset transfer direction in our list of implications at the end of Section 1, and we think that answering these question precisely will require a separate publication.

---

### Official Review · AnonReviewer2 · 2018-11-02
**Highly thought provoking!**

**Rating:** 9
**Confidence:** 4

**Review:**

==== Summary ====

It is widely known that large neural networks can typically be compressed into smaller networks that perform as well as the original network while directly training small networks can be complicated. This paper proposes a conjecture to explain this phenomenon that the authors call “The Lottery Ticket Hypothesis”:  large networks that can be trained successfully contain at initialization time small sub-networks — which are defined by both connectivity and the initial weights that the authors call “winning tickets” — that if trained separately for similar number of iterations could reach the same performance as the large network. The paper follows by proposing a method to find these winning tickets by pruning methods, which are typically used for compressing networks, and then proceed to test this hypothesis on several architectures and tasks. The paper also conjectures that the reason large networks are more straightforward to train is that when randomly initialized large networks have more combinations for subnetworks which makes have a winning ticket more likely.

==== Detailed Review ====

I have found the hypothesis that the paper puts forth to be very appealing, as it articulates the essence of many ideas that have been floating around for quite a while.  For example, the notion that having a large network makes it more probable for some of the initialized weights to be in the “right” direction for the beginning of the training, as mentioned in [1] that was cited in this submission. Given our lack of understanding of the optimization and generalization properties of neural networks, as well as how these two interact, then any insight into this process, like this paper suggests, could have a significant impact on both theory and practice. To that effect, I generally found the experiments in support of the hypothesis to be pretty convincing, or at the very least that there is some truth to it. Most importantly, the hypothesis and experiments presented in this paper gave me a new perspective on both the generalization and optimization problem, which as a theoretician gave me new ideas on how to approach analyzing them rigorously — and that is why I strongly vote for the acceptance of this paper.

Though I have very much enjoyed reading this submission, which for the most part is very well written, it does have some issues:

1. Though this is an empirical paper about an observed phenomenon, it should contain a bit more background and discussion on the theoretical implications of its subject. For example, see [2] which is also an empirical work about a theoretical hypothesis, but still includes the right theoretical context that helps the reader judge the meaning of their results. The same should be done here. For instance, there is a growing interest in the link between compression and generalization that is relevant to this work [3,4], and the effect of winning ticket leading to better generalization could be explained via other works which link structure to inductive bias [5,6].
2. The lottery ticket hypothesis is described in the paper as being both about optimization (faster “convergence”) and about generalization (better “generalization accuracy”). However, there is a slight issue with how these terms are treated in the paper. First, “convergence” is defined as the point at which the test accuracy reaches to a minimum and before it begins to rise again, but it does not mean (and most likely not) that it is the point at which the optimization algorithm converged to its minimum — it is better to write that early stopping regularization was used in this case. Second, the convergence point is chosen according to the test set which is bad methodology, because the test set cannot be used for choosing the final model (only the training and validation sets). Third, the training accuracies are not reported in the paper, and without them, it is difficult to judge if a given model fails to generalize is simply fails to converge to 100% accuracy on the training set. As a minor note, “generalization accuracy” as a term is not that common and might be a bit confusing, so it is better to write “test accuracy”.

To conclude, even though I urge the authors to address the above issues, which could significantly improve its quality and clarity, I think that this article thought-provoking and highly deserving of being accepted to ICLR.

[1] Bengio et al. Convex neural networks. NIPS 2006.
[2] Zhang et al. Understanding deep learning requires rethinking generalization. ICLR 2017.
[3] Arora et al. Stronger generalization bounds for deep nets via a compression approach. ICML 2018.
[4] Zhou et al. Compressibility and Generalization in Large-Scale Deep Learning. Arxiv preprint 2018.
[5] Cohen et al. Inductive Bias of Deep Convolutional Networks through Pooling Geometry. ICLR 2017.
[6] Levine et al. Deep Learning and Quantum Entanglement: Fundamental Connections with Implications to Network Design. ICLR 2018.

==== Updated Review Following Rebuttal ====

The authors have addressed all of the concerns that I have mentioned above, and so I have updated my score accordingly. The additional background on related works, as well as the additional experiments in response to the other reviews will help readers appreciate the observations that are raised by the authors. The new revision is a very strong submission, and I highly recommend accepting it to ICLR.

---

> ### Author Response · Authors · 2018-11-26
> **Author Response**
>
>
> Thank you so much for your thoughtful review. Below, you will find our responses to your questions and comments. We have modified the paper to reflect your feedback, and we are very interested in any further feedback you have about the new version of the paper.
>
> We have summarized the changes in the new version of the paper in a top-level comment called "Summary of Changes in the New Version."
>
> ---
>
> > 1. Though this is an empirical paper about an observed phenomenon, it should contain a bit more background and discussion on the theoretical implications of its subject. For example, see [2] which is also an empirical work about a theoretical hypothesis, but still includes the right theoretical context that helps the reader judge the meaning of their results. The same should be done here. For instance, there is a growing interest in the link between compression and generalization that is relevant to this work [3,4], and the effect of winning ticket leading to better generalization could be explained via other works which link structure to inductive bias [5,6].
>
> We have rewritten our discussion section (Section 6) to connect with contemporary understanding of inductive bias, generalization (and its relation to compressibility), and optimization of overparameterized networks. We hope that this section provides appropriate context for interpreting these results, however we are open to additional suggestions.
>
> ---
>
> > 2. The lottery ticket hypothesis is described in the paper as being both about optimization (faster “convergence”) and about generalization (better “generalization accuracy”). However, there is a slight issue with how these terms are treated in the paper. First, “convergence” is defined as the point at which the test accuracy reaches to a minimum and before it begins to rise again, but it does not mean (and most likely not) that it is the point at which the optimization algorithm converged to its minimum — it is better to write that early stopping regularization was used in this case.
>
> Thank you for this very helpful suggestion. We have updated our language throughout the paper to ensure that we are using this terminology properly.
>
> ---
>
> > Second, the convergence point is chosen according to the test set which is bad methodology, because the test set cannot be used for choosing the final model (only the training and validation sets).
>
> We have updated all of our experiments in the main body of the paper to report the iteration of early-stopping based on validation loss and to report the accuracy at that iteration based on test loss. The conclusions from our results remain the same.
>
> ---
>
> > Third, the training accuracies are not reported in the paper, and without them, it is difficult to judge if a given model fails to generalize is simply fails to converge to 100% accuracy on the training set.
>
> We have updated the paper to include graphs of the training accuracies at early-stopping time for lenet and conv2/4/6. In general, training accuracy at early-stopping time rises with test accuracy. However, at the end of the training process, training accuracy generally reaches 100% for all but the most heavily pruned networks (see the new Appendix B); this is true for both winning tickets and randomly reinitialized networks (although winning tickets generally still reach 100% training accuracy when pruned slightly further (e.g., 3.6% vs. 1.9% for MNIST)). Even so, the accuracy patterns witnessed at early-stopping time remain in place at the end of training: winning tickets see test accuracy improvements and reach higher test accuracy than when randomly reinitialized, indicating that winning tickets indeed generalize better.
>
> ---
>
> > As a minor note, “generalization accuracy” as a term is not that common and might be a bit confusing, so it is better to write “test accuracy”.
>
> We have updated our language to reflect this suggestion.

---

> > ### Comment · AnonReviewer2 · 2018-11-29
> > **Thank you for your response**
> >
> > Thank you for your response, for addressing my previous concerns with the paper, and for taking the additional time for revising your original submission. Please see my updated review above.

---

### Official Review · AnonReviewer1 · 2018-11-05
**interesting conjecture, needs experiments on larger dataset and better presentation and explanation about the result**

**Rating:** 5
**Confidence:** 4

**Review:**

It was believed that sparse architectures generated by pruning are difficult to train from scratch. The authors show that there exist sparse subnetworks that can be trained from scratch with good generalization performance. To explain the difficulty of training pruned networks from scratch or why training needs the overparameterized networks that make pruning necessary,  the authors propose a lottery ticket hypothesis: unpruned, randomly initialized NNs contain subnetworks that can be trained from scratch with similar generalization accuracy.  They also present an algorithm to identify the winning tickets.

The conjecture is interesting and it is still a open question for whether a pruned network can reach the same accuracy when trained from scratch. It may helps to explain why bigger networks are easier to train due to “having more possible subnetworks from which training can recover a winning ticket”. It also shows the importance of both the pruned architecture and the initialization value. Actually another submission (https://openreview.net/forum?id=rJlnB3C5Ym) made the opposite conclusions.

The limitations of this paper are several folds:

- The paper seems a bit preliminary and unfinished.  A lot of notations seems confusing, such as “when pruned to 21%”. The author defines a winning lottery ticket as a sparse subnetwork that can reaching the same performance of the original network when trained from scratch with the “original initialization”. It is quite confusing as there is no definition anywhere about the “original initialization”. It would be clearer if the author can use some math notations.

- As identified by the authors themself, lacking of supporting experiments on large-scale dataset and real-world models. Only MNIST/CIFAR-10 and toy networks like LeNet, Conv2/Conv4/Conv6 are used. The author has done experiments on resnet, I would be better to move it to the main paper.

- There is no explanation about why the “lottery ticket” can perform well when trained with the “original initialization” but not with random initialization. Is it because the original initialization is not far from the pruned solution? Then this is a kind of overting to the obtained solution.

- The other problem is that the implications are not clearly useful without showing any applications. The paper could be stronger if the authors can provide more results to support the applications of this conjecture.

- The authors only explore the sparse networks. Model compression by sparsification has good compression rate, especially for networks with large FC layers. However, the acceleration relies on specific hardware/libraries. It would be more complete if the author can provide experiments on structurally pruned networks, especially for CNNs.

- The x-axis of pruning ratios in Figure 1/4/5 could be uniformly sampled and make the figure easier to read.

Questions:
- Does the winning tickets always exist?
- What is the size of winning tickets for a very thin network? Would it also be less than 10%?


------update----------

I appreciate the author’s efforts on providing detailed response and more experiments. After reading the rebuttal and the revised version, though the paper has been improved, my concerns are not fully addressed to safely accept it.

It can be summarized that there exists a sparse network that can be trained well only provided with certain weight initialization.The winning tickets can only be found via iterative pruning of the trained network. This is a chicken-egg problem and I failed to see how it can improve the network design. It still feels incomplete to me by just providing a hypothesis with limited sets of experiments. The implications are actually the most valuable/attractive part, such as “Improve our theoretical understanding of neural networks”, however, they are very vague with no clear instructions even after accepting this hypothesis. I would expect analysis of the reason behind failure and success. I understand that it could be left for another paper, but the observations/experiments only are not strong enough for confirming the the hypothesis.

Specifically, the experiments are conducted on relatively wide and shallow CNNs. Note that VGG-16/19 and ResNet-18 are designed for ImageNet but not CIFAR-10, which are much wider than normal CIFAR-10 networks, such as ResNet-56. Even “resnet18 has 16x fewer parameters than conv2 and 75x fewer than VGG19”, it is mainly due to the removal of FC layers with average pooling and cannot be claimed as “much thinner” networks. As increasing the wideness usually ease the optimization, and the pruned sparse network still enjoy this property unless significantly pruned. Thus, I still doubt whether the conclusion can hold for much thinner network, i.e., “winning tickets near or below 10-20%, depending on the level of overparameterization of the original network.”

The observation of “winning ticket weights tend to change by a larger amount then weights in the rest of the network” in Figure 19 seems natural and the conjecture of the reason “magnitude-pruning biases the winning tickets we find toward those containing weights that change in the direction of higher magnitude” sounds reasonable. It would be great if the authors can dig into this and make more comparison with the distribution of random weights initialization.

The figures could also be improved and simplified as the lines are hard to read and compare.

---

> ### Public Comment · (anonymous) · 2018-11-14
> **Re. "original initialization"**
>
> I share many of this reviewer's concerns and hope they can be addressed by the authors.
>
> However, I found the point about "original initialization" to be rather pedantic. The majority of the audience will understand "original initialization" to be the values of the weights before any optimization.
>
> While it is possible that some light verbiage would be helpful to clarify, I do not think that "math notations" will help a bit (and in fact may serve to further confuse).
>
> I am not affiliated with the authors in any way.

---

> ### Author Response · Authors · 2018-11-26
> **Author Response (Part 2)**
>
>
> > The authors only explore the sparse networks. Model compression by sparsification has good compression rate, especially for networks with large FC layers. However, the acceleration relies on specific hardware/libraries. It would be more complete if the author can provide experiments on structurally pruned networks, especially for CNNs.
>
> This is a great observation. We agree that structured pruning techniques produce pruned networks that are more amenable to existing software/hardware acceleration techniques. In the limitations section of the updated version (Section 7), we have explicitly noted structured pruning as an opportunity to connect our empirical observations of winning tickets to concrete practice.
>
> ---
>
> > The x-axis of pruning ratios in Figure 1/4/5 could be uniformly sampled and make the figure easier to read.
>
> Done - thank you for the suggestion!
>
> ---
>
> > Does the winning tickets always exist?
>
> Our experiments indicate that winning tickets do seem to exist for the variety of network architectures considered in this paper (and as explicitly scoped by our stated limitations in Section 7 - we acknowledge that we only consider a limited subset of neural network tasks in this paper). However, in the most literal sense, no: winning tickets do not always exist for all datasets and networks. Take, as an example, a minimal dense network for two-way XOR which has two hidden units. If the parameters of the network are initialized to values that give the correct outputs from the very start, then removing any one parameter makes it impossible to reach the same accuracy as the unpruned network.
>
> ---
>
> > What is the size of winning tickets for a very thin network? Would it also be less than 10%?
>
> In the updated version of the paper (Section 5), we have studied several networks that are much thinner than those described in the original version of the paper: VGG16, VGG19, and resnet18. For VGG16 and VGG19, we continue to find winning tickets that are at or less than 10% of the original size of the network. For resnet18 (which has 16x fewer parameters than conv2 and 75x fewer than VGG19), we find winning tickets that are about 15% of the size of the original network. Our results suggest that, for several exemplary thin networks, we still find winning tickets near or below 10-20%, depending on the level of overparameterization of the original network.

---

> ### Author Response · Authors · 2018-11-26
> **Author Response (Part 1)**
>
> (Edit: we reworded this comment for clarity, but the content is otherwise the same)
>
> Thank you so much for your thoughtful review. Below, you will find our responses to your questions and comments. We have modified the paper to reflect your feedback, and we are very interested in any further feedback you have about the new version of the paper.
>
> We have summarized the changes in the new version of the paper in a top-level comment called "Summary of Changes in the New Version."
>
> Where multiple reviewers made similar comments, we have grouped the answers into a "Common Questions" comment; you can find this comment as a response to our top-level comment called "Summary of Changes in the New Version."
>
> ---
>
> > Actually another submission (https://openreview.net/forum?id=rJlnB3C5Ym) made the opposite conclusions.
>
> Up to a certain level of pruning, a randomly reinitialized network can match the accuracy (and often learning speed) of the original network. We find this to be true throughout our paper, particularly in the conv2/4/6 experiments. However, past this point, winning tickets continue to match the performance of the original network when randomly reinitialized networks cannot. Furthermore, at the levels of pruning for which randomly reinitialized networks do match the performance of the original network, winning tickets reach even higher accuracy and learn faster. As a concrete example, in Section 5 of the updated version of our paper, we include lottery ticket experiments on the same VGG19 network for CIFAR10 as appears in "Rethinking the Value of Network Pruning." We find that, when randomly reinitialized, subnetworks found via iterative pruning remain within 0.5 percentage points of the accuracy of the original network until pruned by about 70%; after this point, accuracy drops off as in random reinitialization experiments throughout our paper. This result supports the findings of "Rethinking the Value of Network Pruning:" up to a certain level of pruning, VGG19 continues to reach accuracy close the original network even when randomly reinitialized. However, past the initial two or three pruning iterations, these randomly reinitialized networks do not qualify as winning tickets by our definition. In contrast, iterative pruning produces winning tickets when the network is pruned by up to 94.5%.
>
> ---
>
> > It would be clearer if the author can use some math notations.
>
> We agree; thank you for the feedback. In the updated version, we have made our definitions precise through mathematical notation.
>
> ---
>
> > As identified by the authors themself, lacking of supporting experiments on large-scale dataset and real-world models. Only MNIST/CIFAR-10 and toy networks like LeNet, Conv2/Conv4/Conv6 are used. The author has done experiments on resnet, I would be better to move it to the main paper.
>
> Please see "Common Questions."
>
> ---
>
> > There is no explanation about why the “lottery ticket” can perform well when trained with the “original initialization” but not with random initialization. Is it because the original initialization is not far from the pruned solution? Then this is a kind of overting to the obtained solution.
>
> Please see "Common Questions."
>
> ---
>
> > The other problem is that the implications are not clearly useful without showing any applications. The paper could be stronger if the authors can provide more results to support the applications of this conjecture.
>
> We largely consider the value of this paper to be its identification of an avenue to understand properties of neural networks, independent of the current applicability of this understanding to end objectives (e.g., faster training). We intend for this paper to pose an opportunity for future applications. However, we agree that we do not evaluate them.
>
> If winning tickets do seem to exist in a wide variety of networks, we believe that the most concrete application is in line with contemporary work on distillation/compression/pruning: if a technique can find winning tickets early on in training, then those winning tickets can be used for the remainder of learning, thereby reducing resource demands and speeding up learning (depending on the profitablity of exploiting the sparsity of a winning ticket, as you note next).

---

### Author Response · Authors · 2018-11-26
**Summary of Changes in New Version**

(Edit: we reworded this comment for clarity, but the content is otherwise the same)

We would like to thank the reviewers for their thorough feedback. In response to the many valuable suggestions and questions they provided, we have made substantial revisions to the paper. In this comment, we summarize those changes section-by-section.

-----

Changes throughout the paper:

* As suggested by Reviewer 2, we no longer refer to network "convergence." Instead, we describe the same phenomenon as "the iteration at which early-stopping would occur."  Rather than discussing faster convergence times, we instead refer to faster learning as indicated by an earlier iteration of early-stopping.

* As suggested by Reviewer 1, we have added mathematical notation throughout the paper where appropriate. We adopt the syntax P_m = k% to describe a winning ticket for which the pruning mask m contains 1's in k% of its indices.

* As suggested by Reviewer 2: for all of our training iterations/test accuracy experiments, we measure early-stopping with the validation set and report accuracy at early-stopping using the test set. Our results throughout the paper are the same as in the original submission.

-----


Section 1:

* As suggested by Reviewer 1, we have added a formal characterization of the lottery ticket hypothesis in mathematical notion.  The meaning of this statement is the same as the informal statement made in the original submission.

-----

Section 2:

* As suggested by Reviewer 2, we have added graphs that show training accuracy at early-stopping time and test accuracy at the end of training (i.e., when training accuracy reaches 100%). Generating this data required re-running our experiments. Therefore, we have updated all reported numbers in this section to reflect the recollected values. Our results remain the same.

* We integrated the P_m notation to streamline the prose. Otherwise,the semantics of this text is exactly the same.

-----

Section 3: We applied the same changes as in Section 2 (described above). Our results remain the same.

-----

Section 4: This section compares results with dropout to results from Section 3. The only change we make is an update to the numbers reported from Section 3 (which are updated as described above). Otherwise, our results are the same.

-----

Section 5: As suggested by Reviewers 1 and 3, we have moved the content for resnet-18 on CIFAR 10 that was in Appendix D in the original submission to this section. Additionally, we provide new experiments for VGG16/19 on CIFAR10.

To briefly summarize our results, we continue to find winning tickets. However, we show that our results are sensitive to learning rate (as was previously reported for resnet-18 in Appendix D in the original submission). Specifically, at the higher learning rates typically used to train these networks, there is a small accuracy gap between the identified winning ticket and the original network. We show that learning rate warmup eliminates this gap.

-----

Section 6: As suggested by Reviewer 2, we have expanded this section to integrate theoretical context related to generalization, optimization, and inductive bias. Otherwise, our conclusions remain the same.

-----

Section 7: We have added content to our Limitations to reflect the additions that we have promised in our responses to individual reviews.

-----

Section 8: Unchanged.

-----

Appendices:

We have added content to our Appendix to reflect the additions that we have promised in our responses to individual reviews.

---

> ### Author Response · Authors · 2018-11-26
> **Responses to Common Reviewer Questions**
>
> There were a couple of questions that were asked by more than one reviewer. We have centralized our responses to those common questions here.
>
> ---
>
> > Reviewer 1: As identified by the authors themself, lacking of supporting experiments on large-scale dataset and real-world models. Only MNIST/CIFAR-10 and toy networks like LeNet, Conv2/Conv4/Conv6 are used. The author has done experiments on resnet, I would be better to move it to the main paper.
>
> > Reviwer 3: The paper would be stronger, if the existence of winning tickets on larger-scale experiments or tasks other than classification were shown - even if these experiments did not have a large number of control experiments/ablation studies.
>
> In the new version of the paper, we have added experiments on resnet18 and vgg16/19 with CIFAR10 (Section 5 for VGG19 and resnet-18 and Appendix H for VGG16), where we continue to find winning tickets. Notably, our iterative-pruning method for finding winning tickets becomes sensitive to learning rate, so we have to modify the learning rate schedule from the default values to find winning tickets (e.g., by adding warmup). Unfortunately, running pruning experiments on Imagenet or the like was beyond our means during the rebuttal period. The new experiments, which better evoke real-world architectures, improve our confidence in the generality of the lottery ticket hypothesis. However, we acknowledge this concern.
>
> ---
>
> > Reviewer 1: There is no explanation about why the “lottery ticket” can perform well when trained with the “original initialization” but not with random initialization. Is it because the original initialization is not far from the pruned solution? Then this is a kind of overting to the obtained solution.
>
> > Reviewer 3: While the paper shows the existence of winning tickets robustly and convincingly on the networks/tasks investigated, the next important question would be how to systematically and reliably “break” the existence of lottery tickets. Can they be attributed to a few fundamental factors?
>
> We have not yet been able to definitively answer why a winning ticket can perform well with the original initialization but not random initialization. However, in the updated version, we have added an appendix that provides more detail about the internals of winning tickets from lenet for MNIST (Appendix D). Specifically, we investigate two questions: 1) (as suggested by Reviewer 1) are the initial values of winning tickets close to their trained values? and 2) what is the distribution of weights in winning tickets before initialization?
>
> * Question 1: we actually find the opposite of what Reviewer 1 suggests: in the unpruned network, weights that are part of the eventual winning tickets tend to move more than weights that are not part of the winning ticket.
>
> * Question 2: we find that the winning ticket initializations tend to come from a different distribution than the network as a whole: a bimodal distribution with two peaks toward the extremes of the truncated normal distribution from which the network was originally initialized. We try reinitializing winning tickets from this distribution, but doing so performs no better than random reinitialization. We also try performing magnitude pruning before training based on the hypothesis that low-magnitude weights are unlikely to be part of the eventual winning ticket; this approach also performs no better than random reinitialization. We conclude that these insights based on magnitude at initialization are not sufficient to identify a lottery ticket.
>
> These results do not definitively answer the questions posed, but they represent the first set of clues on the path to doing so. We intend to continue down this path in our future work.

---

### Public Comment · (anonymous) · 2018-12-04
**Winning tickets obtained by a different pruning method (SNIP) and the effect of re-initialization**

Thank you for the interesting work.

Concurrently, we proposed a new pruning method, SNIP, ( https://openreview.net/forum?id=B1VZqjAcYX ), that finds extremely sparse networks by single-shot at random initialization, and the pruned sparse networks are then trained in the standard way.

We found one of your hypotheses "When randomly reinitialized, a winning ticket learns more slowly and achieves lower test accuracy" intriguing. Therefore, we tested to see if this behavior holds on subnetworks obtained by SNIP.

Specifically, we tested various models (LeNets, AlexNets, VGGs and WRNs) on MNIST and CIFAR-10 datasets for the same extreme sparsity levels (> 90%) used in our paper. As a result, we found that there are no differences in performance between re-initializing and NOT-initializing the subnetworks (after pruning by SNIP and before the start of training): 1) the final accuracies are almost the same (the difference is less than 0.1%) and 2) the training behavior (the training loss and validation accuracy curves) is very similar.

It seems that our finding, albeit preliminary, is contradictory to the aforementioned hypothesis. This discrepancy may be due to the fact that the conclusions in your paper are based on magnitude based pruning and the method is tested for moderate sparsity levels, etc.

As stated in your latest version (Section 7), "we intend to explore more efficient methods for finding winning tickets that will make it possible to study the lottery ticket hypothesis in more resource-intensive settings" or "... non-magnitude pruning methods (which could produce smaller winning tickets or find them earlier)", we believe that SNIP could be a method of choice for the further exploration of your hypotheses.

We hope to hear your thoughts.

---

> ### Author Response · Authors · 2018-12-05
> **Thank you for sharing your work!**
>
> (Edited to improve clarity and update replication results.)
>
> Thank you for sharing your work; we are very excited to see your results, since they seem to support the lottery ticket hypothesis as posed and add substantial further evidence to our hypothesis via a different pruning technique. We will be sure to refer to the SNIP results in the final version of our paper.
>
> The main statement of the lottery ticket hypothesis does not exclude the possibility that winning tickets are still trainable when reinitialized. Specifically, while the hypothesis conjectures that, given a dense network and its initialization, there exists a subnetwork that is still trainable with the original initializations, it does not require any particular behavior of this subnetwork under other initializations. Thank you for this comment; we will revise our language to make this clear.
>
> In our experiments, we do find initialization to have a significant impact on the success of the pruned subnetworks we find (hence the quote you provide from our paper).  You mention in your rebuttal that “SNIP finds the architecturally important parameters in the network,” perhaps reducing the relative importance of initialization for the winning tickets that you find.
>
> Once your source code is made available, we would be very interested in analyzing your preliminary comparison between SNIP-pruned networks with the original initialization and SNIP-pruned networks when reinitialized; we have replicated the SNIP algorithm as presented in your paper in our own framework and produce the following results:
>
> * Lenet (MNIST): We confirm that the accuracy of SNIP-pruned networks does not change when they are reinitialized. In addition, we find that, although SNIP outperforms random pruning, SNIP-pruned networks do not match the test accuracy of our winning tickets or our randomly reinitialized winning tickets.
>
> * Resnet-18 (CIFAR10):  We did not have time (in the 24 hours between your comment and the end of the comment period) to confirm your random reinitialization experiments on this network. We find that, although SNIP outperforms random pruning, it does not match the test accuracy of the winning tickets and only slightly outperforms the randomly-reinitialized winning tickets.
>
> * VGG19 (CIFAR10): (Updated) We confirm that the accuracy of the SNIP-pruned networks does not change when they are reinitialized. When training with warmup, SNIP produces networks that nearly match the accuracy of our winning tickets at the corresponding level of sparsity. However, our winning tickets learn faster than the SNIP-pruned networks.
>
> We look forward to discussing SNIP in the final version of our paper as a potential “method of choice for the further exploration of [our] hypotheses.” However, our preliminary, replicated results suggest that there is a gap in accuracy and speed of learning between SNIP-pruned networks and our winning tickets.
>
> (One minor nit: you mention that we only test our method for moderate sparsity levels, but our graphs show that we continue to find winning tickets at extreme sparsity levels (> 90%) similar to those in your paper.)

---

> > ### Public Comment · (anonymous) · 2018-12-06
> > **Thank you for your response.**
> >
> > We appreciate that the authors spared time to address our comment, and we believe that the confusion on the effect of (re-)initialization is clarified.
> > We look forward to trying SNIP in your experimental setting once your code is released.

---

> > > ### Author Response · Authors · 2018-12-12
> > > **Additional Experiments and Graphs**
> > >
> > >
> > >
> > > We have an update with several further experiments that examine the relationship between SNIP and our paper.
> > >
> > > We have simplified our pruning mechanism to prune weights globally (instead of per-layer) with otherwise the same pruning technique. For our three main networks (MNIST, Resnet-18, and VGG-19), we find that globally-pruned winning tickets reach higher accuracy at higher levels of sparsity and learn faster than SNIP-pruned networks.
> > >
> > > For example, VGG19 reaches 92% test accuracy when pruned by at most 97.2% with SNIP vs. at most 99.5% for globally-pruned winning tickets. Resnet-18 achieves 90% accuracy when pruned by at most 27% with SNIP vs. at most 89% for globally-pruned winning tickets.
> > >
> > > We also performed several further experiments exploring the effect of initialization and structure on SNIP-pruned networks. We find that SNIP-pruned networks can be randomly reinitialized as well as randomly rearranged (i.e., randomly choose the locations of unpruned connections within layers) with limited impact on their accuracy. However, these networks are neither as accurate nor learn as quickly as winning tickets.
> > >
> > > The fact that SNIP-pruned networks can be rearranged suggests that SNIP largely identifies the proportions in which layers can be pruned such that the network is still able to learn, leaving significant opportunity to exploit the additional, initialization-sensitive understanding demonstrated by our results.
> > >
> > > We provide several graphs here (https://drive.google.com/drive/folders/1lpxJFpkF0Afq1rRqkEDnLcPN0kMV8BBC?usp=sharing) to support these claims. We will add these experiments to the final version of our paper.

---

### Public Comment · ~Kevin_Martin_Jose1 · 2019-06-02
**Minor typo**

Nothing of consequence, I just found a typo in this sentence:

"This work was support in part by the Office of Naval Research (ONR N00014-17-1-2699)"

should be

"This work was supported in part by the Office of Naval Research (ONR N00014-17-1-2699)"

Great paper BTW.

---

### Public Comment · ~Hady_Elsahar2 · 2019-10-14
**Why the winning ticket training accuracy increases while iterative pruning?**

Figure 5 Iterative pruning showing training scores after iterative pruning.

The conv-2 (blue solid line) training accuracy increases ~10% when iteratively pruned until when only 7% of the weights are retained the model starts not fitting the data anymore. This effect happens less for large models conv-4 and conv-6

Models fit slightly faster as well, that is a bit counterintuitive to me is there a justification to that?

---

### Public Comment · ~Rahmawati_Pratiwi1 · 2020-06-20
**Study Case**

Very interesting theory indeed. I wonder if you could make a study case comparing this method to lottery from other country such as Indonesia? As I think other countries have different system. That would make it much more interesting in my opinion.

---

### Meta-Review · Area_Chair1 · 2018-12-12
**Intriguing hypothesis with convincing experimental validation and analyses**

**Confidence:** 4
**Recommendation:** Accept (Oral)

**Metareview:**

The authors posit and investigate a hypothesis -- the “lottery ticket hypothesis” -- which aims to explain why overparameterized neural networks are easier to train than their sparse counterparts. Under this hypothesis, randomly initialized dense networks are easier to train because they contain a larger number of “winning tickets”.
This paper received very favorable reviews, though there were some notable points of concern. The reviewers and the AC appreciated the detailed and careful experimentation and analysis. However, there were a couple of points of concern raised by the reviewers: 1) the lack of experiments conducted on large-scale tasks and models, and 2) the lack of a clear application of the idea beyond what has been proposed previously.

Overall, this is a very interesting paper with convincing experimental validation and as such the AC is happy to accept the work.